

# On the time and length scales of the Arctic sea ice thickness anomalies: a study based on fourteen reanalyses

Leandro Ponsoni[1], François Massonnet[1], Thierry Fichefet[1], Matthieu Chevallier [2], and David Docquier[1]

[1]Georges Lemaître Centre for Earth and Climate Research (TECLIM), Earth and Life Institute, Université catholique de Louvain, Louvain-la-Neuve, Belgium
[2]Centre National de Recherches Météorologiques (CNRM), Météo France/CNRS UMR3589, Toulouse, France

*Correspondence to:* Leandro Ponsoni (leandro.ponsoni@uclouvain.be)

**Abstract.** The ocean–sea ice reanalyses are one of the main sources of Arctic sea ice thickness data both in terms of spatial and temporal resolution, since observations are still sparse in time and space. In this work, we first aim at comparing how the sea ice thickness from an ensemble of fourteen reanalyses compares with different sources of observations, such as moored upward-looking sonars, submarines, airbornes, satellites and ice boreholes. Second, based on the same reanalyses, we intent to

characterize the time (persistence) and length scales of sea ice thickness anomalies. We investigate whether data assimilation of sea ice concentration by the reanalyses impacts the realism of sea ice thickness as well as its respective time and length scales. The results suggest that reanalyses with sea ice data assimilation do not necessarily perform better in terms of sea ice thickness compared with the reanalyses which do not assimilate sea ice concentration. However, data assimilation has a clear impact on the time and length scales: reanalyses built with sea ice data assimilation present shorter time and length scales. The

mean time and length scales for reanalyses with data assimilation vary from 2.5–5.0 months and 337.0–732.5 km, respectively, while reanalyses with no data assimilation are characterized by values from 4.9–7.8 months and 846.7–935.7 km, respectively.

## 1   Introduction

The variability of the Arctic sea ice has received increasing attention from the scientific community over the past years (e.g., Chevallier and Salas-Mélia, 2012; Stroeve et al., 2014; Blanchard-Wrigglesworth and Bitz, 2014; Guemas et al., 2016). The

main reason lies in the fact that Arctic sea ice plays a key role in the Earth's climate system (Budyko, 1969; Manabe and Stouffer, 1980b; Maykut, 1982). Among other contributions, it has been suggested that a decline of the Arctic sea ice extent and volume leads to a weakening of the Atlantic Meridional Overturning Circulation (Sévellec et al., 2017) and, therefore, potentially impacts the global distribution of heat (Drijfhout, 2015; Hansen et al., 2016). At the same time, the Arctic is one of the most sensitive regions to climate changes due to a phenomenon known as Arctic amplification (Manabe and Stouffer,

1980a; Holland and Bitz, 2003; Serreze et al., 2009). For instance, the current observed warming in the Arctic is reported to be nearly twice as large as other regions of the globe (Anisimov et al., 2007).

    Other multiple specific interests from different stakeholders have reinforced the importance of the sea ice projections, both at regional and larger scales, that include: shorter shipping lanes (Lindstad et al., 2016), travel and tourism industry (Handorf, 2011), hunting and fishing activities (Nuttall et al., 2005), mineral resource extraction (Gleick, 1989), potential impact on the



weather at mid-latitudes (Walsh, 2014), environmental hazards (Nelson et al., 2002) and loss of weather predictive power by indigenous communities (Krupnik and Jolly, 2002). In this context, the sea ice thickness (SIT) is likely the most relevant state variable for monitoring, forecasting and understanding both recent and future changes in terms of the Arctic sea ice. First, because this parameter provides predictive information for the sea ice extent anomalies (Lindsay et al., 2008; Holland et al.,

2011) and, second, due to the fact that SIT anomalies persist longer than sea ice extent anomalies, the former being reported as a forcing of the latter (Blanchard-Wrigglesworth et al., 2011).

However, direct observations of SIT and/or related parameters, namely draft and freeboard, are still sparse in time and space, besides the continuous efforts for compiling former and recent datasets from a range of sources (Lindsay, 2010; Lindsay and Schweiger, 2015). Some recent observational programmes, such as the Year Of Polar Prediction (YOPP) (Jung et al.,

2016) and the MOSAiC International Arctic Drift Expedition (http://www.mosaicobservatory.org/), aim to enhance the Arctic observational system, being especially useful for improving our modeling and forecasting skills.

Due to this lack of direct measurements in the past and present-day, the ocean–ice reanalyses deserve special attention. A reanalysis product consists of models' outputs, which are generated over a certain time span by the same model, configurations and procedures, and so distributed onto regular grids, evenly stepped in time. These products are often built with assimilation

of observational dataset(s) in order to improve the estimate of a certain parameter. For instance, SIT is often estimated by assimilating atmospheric, oceanic and, eventually, sea ice concentration data. The ocean–ice reanalyses are likely the main and more robust source of SIT data in terms of spatio-temporal resolution, being also broadly used for initialization and assimilation in other climate models (e.g., Guemas et al., 2016). Additionally, long-term reanalyses are crucial for understanding the past Arctic sea ice characteristics, in a period when *in situ* observations of ice parameters were inexistent.

In this work we make use of fourteen state-of-the-art reanalyses in order to study two important aspects of the SIT predictability: the time scale (or persistence) and the length scale of SIT anomalies (see Sec. 2.3). Their importance is reinforced by the fact that the predictability of the SIT field depends on how long the anomalies persist over time and on the spatial scale of the respective anomalies. Notice that, hereafter, besides the traditional definition of time predictability, we adopt this term also for the spatial scale. In addition, time and length scales may also be useful for designing an optimal observation system,

when selecting ideal locations for deploying instruments, as well as for defining a frequency sampling strategy (Blanchard-Wrigglesworth and Bitz, 2014).

Blanchard-Wrigglesworth and Bitz (2014) reported SIT anomalies with typical time and length scales of about 6–20 months and 500–1000 km, respectively. These results reinforce the fact that SIT anomaly persists longer compared to sea ice area anomaly, which is reported with a time scale of 2–5 months (Blanchard-Wrigglesworth et al., 2011; Day et al., 2014).

Blanchard-Wrigglesworth and Bitz (2014) suggested that the SIT anomalies from models characterized by a thinner mean ice state tend to present shorter persistence, but larger spatial scales. Blanchard-Wrigglesworth et al. (2011) reported a decline in the time scale of sea ice volume anomalies, as a result of the ice thinning induced by the recent climate changes.

The first aim of this study is to evaluate the performance of different reanalysis products regarding their SIT realism by comparing these reanalyses against observational datasets. A point of main interest is to identify whether or not the assimilation

of sea ice concentration by the reanalyses improves the representation of SIT. Second, we seek to characterize the time and



length scales of the Arctic SIT anomalies. Again, we verify whether or not sea ice data assimilation plays a role in the temporal and spatial scales of SIT anomalies. Furthermore, we investigate the long-term evolution of time and length scales, as well as the relationship between these two parameters.

The manuscript is organized as follows: Section 2 introduces the reanalysis products, the observational datasets and the respective methods applied in this research; Section 3 compiles all results, including the comparison between observations and reanalyses (Section 3.1), the comparison of reanalyses themselves (Section 3.2), and the patterns of time (Section 3.3) and length (Section 3.4) scales. Lastly, Section 4 draws discussion and conclusions on the findings reported in the previous sections.

## 2 Data and methods

### 2.1 Sea ice reanalyses

Monthly fields of SIT from fourteen state-of-the-art ocean–ice reanalyses are used in this work. They were all but one compiled in the context of the ORA-IP project (Balmaseda et al., 2015; Chevallier et al., 2017; Uotila et al., 2018). The ORA-IP reanalyses (and their respective author/provider institution) are: C-GLORS05 (CMCC, Storto et al. (2014)), ECCO-v4 (JPL/NASA, MIT, AER, Forget et al. (2015)), ECDA (GFDL/NOAA, Zhang et al. (2013); Chang et al. (2013)), G2V3 (Mercartor Océan, Ferry et al. (2010)), GECCO2 (University of Hamburg, Köhl (2015)), GloSea5 (UK Met Office, Blockley et al. (2014)), GloSea5-GO5 (UK Met Office, Megann et al. (2014)), MERRA-Ocean (GSFC/NASA/GMAO, Rienecker et al. (2011)), MOVE-CORE (MRI/JMA, Danabasoglu et al. (2014)), MOVE-G2 (MRI/JMA, Toyoda et al. (2013)), ORAP5 (ECMWF, Zuo et al. (2015); Tietsche et al. (2017)), TOPAZ4 (TP4, ARC MFC, Sakov et al. (2012)) and UR025-4 (University of Reading, Valdivieso et al. (2014)). The fourteenth reanalysis is the Pan-Arctic Ice-Ocean Modeling and Assimilation System – PIOMAS (Zhang and Rothrock, 2003). For the acronyms, the reader is referred to Appendix A. The original horizontal grids range from $0.25°$ to $1°$. For comparison, all reanalyses are interpolated onto a common grid of $1° \times 1°$ spatial resolution (Chevallier et al., 2017).

Specific characteristics of each reanalysis regarding horizontal resolution, ocean–sea ice model, spanning period, data output frequency, atmospheric forcing data, type of vertical discretization, subgrid-scale ice thickness distribution, ice dynamics (*VP* viscous-plastic or *EVP* elastic-viscous-plastic), parameters for the ice strength formulation, air-ice drag coefficient, ocean-ice drag coefficient and the presence (and respective method) of ice data assimilation were described in details by Chevallier et al. (2017) (their Table 1) and Balmaseda et al. (2015). For PIOMAS, a complete description is provided by Zhang and Rothrock (2003).

### 2.2 Observational references

We use a compilation of sixteen observational datasets available in the Unified Sea Ice Thickness Climate Data Record (Sea Ice CDR; Lindsay (2010) – http://psc.apl.uw.edu/sea_ice_cdr). The Sea Ice CDR is a concerted effort to bring together a range




of datasets in a consistent format, but originally sampled by different methods and spatio-temporal scales, as well as stored in a variety of formats. We use the post-processed version of the Sea Ice CDR data, that is distributed in monthly mean for moored upward-looking sonar (ULS) ponctual measurements or 50-km averages for submarine, airborne and satellite observations.

From these sixteen datasets, eleven provide draft measurements, while the remaining five provide sea ice thickness data.
Seven draft datasets were sampled by means of moored ULSs, namely "North Pole Environmental Observatory" (NPEO; Drucker et al. (2003); Rothrock and Wensnahan (2007)), "Beaufort Gyre Exploration Project" (BGEP), "Institute of Ocean Science (IOS) - Eastern Beaufort Sea" (IOS-EBS) and "- Chuck Sea (IOS-CHK)", "Alfred Wegener Institute - Greenland Sea" (AWI-GS; Harms et al. (2001)), "Bedford Institute of Oceanography Lancaster Sound" (BIO-LS; Pettipas et al. (2008); Prinsenberg and Pettipas (2008); Prinsenberg et al. (2009)), and "Polar Science Center - Davis Strait" (Davis_St; Drucker et al. (2003)). Four other draft datasets are also based on ULS measurements, but installed on US and UK submarines: "US Navy Submarines - Analog" (US-Subs-AN), "US Navy Submarines - Digital" (US-Subs-DG; Tucker III et al. (2001); Wensnahan and Rothrock (2005); Rothrock and Wensnahan (2007)), "UK Navy Submarines - Analog" (UK-Subs-AN), and "UK Navy Submarines - Digital" (UK-Subs-DG; Wadhams and Horne (1980); Wadhams (1984)).

From the ensemble of sea ice thickness datasets, the "Ice Thickness Program" ran by Environmental Canada (CanCoast) is the only dataset providing direct measurement of ice thickness by means of ice boreholes. The "NASA Operation IceBridge datasets" (IceBridge-V2 and IceBridge-QL; Kurtz et al. (2013)) are derived from aircraft-mounted laser altimeter. Finally, two datasets come from satellite campaigns: the laser-altimeter derive "ICESat Mission-Goddard" (ICESat1-G; Zwally et al. (2008)) from National Aeronautics and Space Administration (NASA) and the radar altimeter-derived "CryoSat satellite data" (CryoSat-AWI; Ricker et al. (2014)) from the European Space Agency (ESA).

## 2.3 Methods

Reanalyses are compared against observations by selecting SIT values from the nearest grid points to the respective observational sites, during the same respective months. Complementary metrics are employed to evaluate the relationship between both datasets. When directly comparing SIT from reanalyses and observations, we estimate the Root Mean Squared Error (RMSE), the correlation coefficient (R) and the Mean Residual Sum of Squares (MRSS) from the linear fit between both datasets, by having the reanalysis values as predictors and the observational values as predicted variables. When comparing SIT from reanalyses with draft from observations, we estimate R and MRSS. Though SIT and draft are different variables, we consider the linear relationship between both parameters given by the hydrostatic equation, but we do not account for snow variation in order to avoid adding uncertainties. RMSE and R are also used for a comparison among all pairs of reanalyses.

SIT anomalies are derived by eliminating the trend and the seasonal cycle present in the time series. To do so, the trend is estimated separately for every month (Jan, Feb, ..., Dec) by means of a 2nd-order polynomial fit and subtracted from the respective month. A 2nd-order fit seems to better reproduce the trends when compared to the linear fit, although the results are very similar (not shown). The same method is applied for the analyses conducted with the pan-Arctic sea ice volume anomaly, derived from the SIT data, as illustrated in Fig. 1.





**Figure 1.** Sea ice volume anomalies estimated from all reanalyses. Anomalies are calculated by excluding the trend and the seasonal cycle. Reanalyses labeled in blue and red highlight whether the datasets were built with or without sea ice data assimilation, respectively.





Grid point comparisons of SIT anomalies among all reanalyses are quantified by RMSEs calculated over an overlapping period of 15 years, from Jan/1993 to Dec/2007. This time span corresponds to the period during which data is available from all reanalyses. Furthermore, as adopted by Blanchard-Wrigglesworth and Bitz (2014), only grid points wherein the mean ice thickness at the time of summer minimum is greater than 0.1 m are taken into account. This condition is valid for all reanalysis-based results, unless otherwise stated.

The time scale (or persistence) is derived from individual time series by calculating the lagged autocorrelation stepped forward by one measurement, equivalent to one month. The $e$-folding reference is used so that the persistence is assumed to be the time when the lagged autocorrelation curve crosses the $1/e$ ($\sim 0.3679$) value, as proposed in previous works (e.g., Blanchard-Wrigglesworth and Bitz, 2014; Guemas et al., 2016). As an example, Fig. 2 displays the time scale derived from the mooring-based draft anomaly sampled in the framework of the BGEP at 150°W, 75°N, from Aug/2003 to Aug/2013. Fig. 2 also shows the time scales from the allocated reanalysis-based SIT anomalies. For this geographical location and time span, the draft anomaly from BGEP persists for about 3.7 months, while the SIT anomalies from the different reanalyses persist from 2.4 to 8 months. The persistence is estimated both from a regional and pan-Arctic perspective. First, it is calculated at each grid point, for all SIT anomaly time series. Second, it is estimated for the long-term (GECCO2 and MOVE-CORE) pan-Arctic ice volume anomalies. For the latter case, we evaluate how stable the $e$-folding time scale is over time by applying a moving (stepped by 1 month) and length-variable window (from 5 to 59 years). Here, we also investigate whether the moving time scale is marked by significant band(s) of variability. To do so, we applied wavelet analysis as proposed by Torrence and Compo (1998).

The length scales of the SIT anomalies are estimated for the reanalysis datasets. The first step is to determine one-point correlation maps. In other words, we calculate the cross-correlation between the SIT anomaly from each grid point with the anomaly from all other points. Subsequently, we make use of the $e$-folding reference and, for every map, we select all grid cells with correlation coefficient higher than $1/e$. The radius of a circle that yields the area covered by these selected cells is defined as the length scale of the SIT anomaly. This methodology is detailed and graphically presented by Blanchard-Wrigglesworth and Bitz (2014). Fig. 3a shows an example where the length scale is calculated for the SIT anomalies from CryoSat seasonal data: Spring (Mar-Apr) and Autumn (Oct-Nov), from Autumn 2010 to Spring 2017. In turn, Fig. 3b-c reveal that a similar length scale pattern is also present in PIOMAS. It is worthwhile mentioning that this illustrative example allows a first assessment on how length scales from observations and reanalyses compare to each other. However, it can not be compared to the spatial scales of monthly anomalies further studied in Sec. 3.4.

## 3 Results

### 3.1 Comparison of reanalyses with observations

The scatter plots shown in Fig. 4 combine SIT from each reanalysis and the observational datasets from all sources. The latter are separated into two parameters: draft (black dots) and SIT (green dots). The comparisons indicate that all reanalyses are significantly correlated to the observations, whether these are draft or SIT. By comparing SIT and draft, four reanalyses have



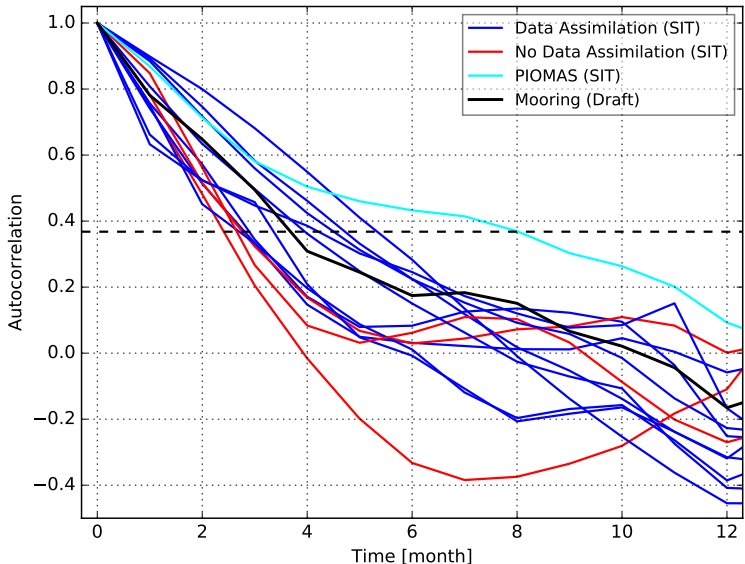

**Figure 2.** Autocorrelation curves for the draft time series sampled *in situ* by upward-looking sonars deployed in the BGEP oceanographic mooring (black line). This mooring was placed at 150°W, 75°N (see location in Fig. 3c) and the data spans from Aug/2003 to Aug/2013. The blue and red lines display the autocorrelation estimated from the SIT anomalies time series for the ORA-IP reanalyses, at the nearest grid point to the mooring and same time span, for the reanalyses built with and without sea ice data assimilation, respectively. The cyan line indicates the autocorrelation estimated for the PIOMAS reanalysis. The time in which the curves cross the black dashed line is defined as their respective *e*-folding time scales.

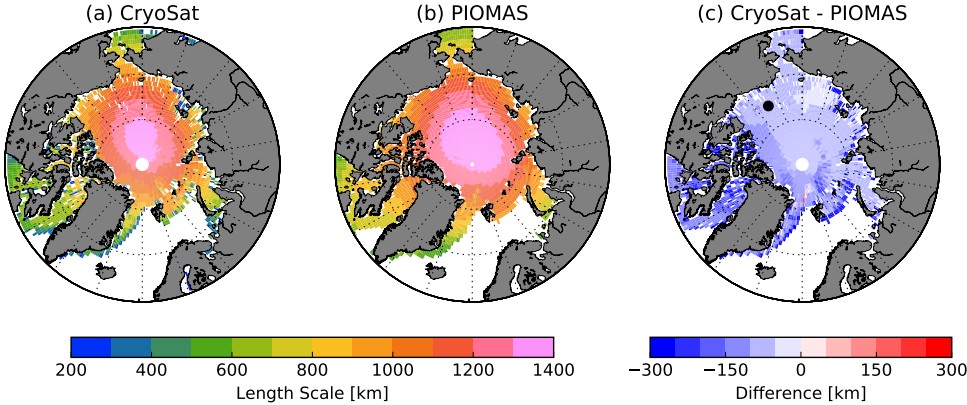

**Figure 3.** (a) E-folding length scale estimated from the CryoSat seasonal data of sea ice thickness. This dataset contains fourteen Spring (Mar-Apr) and Autumn (Oct-Nov) fields, starting in Autumn 2010 and finishing in Spring 2017. (b) Same as (a), but using the equivalent temporal averages from the PIOMAS data. The difference between the fields shown in (a) and (b) is plotted in (c). The black circle in (c) indicates the location of the mooring from which data is used in Fig. 2.

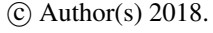


correlation coefficients larger than 0.7: TP4 (R = 0.76), C-GLORS05 (0.74), MOVE-CORE (0.74) and UR025-4 (0.73). On the other hand, GECCO2 (0.17) and MOVE-G2 (0.10) are marked by the weakest correlations. If we evaluate the reanalyses' statistical capability for predicting the observational values, the MRSS from the linear fit indicates that TP4 (MRSS = 0.39 $m^2$), UR025-4 (0.42 $m^2$) and C-GLORS05 (0.49 $m^2$) are the best predictors, while MERRA-OCEAN (1.42 $m^2$) and GECCO2

(1.27 $m^2$) provide the lowest agreement.

When comparing SIT from both datasets, the reanalyses with higher correlation coefficient are PIOMAS (R = 0.66), GECCO2 (0.64) and TP4 (0.61), while ECDA (0.43), ECCO-v4 (0.40) and MOVE-G2 (0.30) are the reanalyses with poorest correlation. In terms of linear fit, PIOMAS (MRSS = 0.41 $m^2$), TP4 (0.41 $m^2$), GECCO2 (0.42 $m^2$), ORAP5 (0.46 $m^2$) and C-CGLORS05 (0.49 $m^2$) are the best performing predictors (MRSS < 0.5 $m^2$), while MOVE-CORE (0.71 $m^2$) and ECCO-v4

(0.7 $m^2$) provide the lowest prediction capability. In addition, a direct comparison by means of RMSEs indicates which reanalyses are closer to the ensemble of observations, as follows: PIOMAS (RMSE = 0.7 m), C-GLORS05 (0.8 m), GloSea5-GO5 (0.8 m), ORAP5 (0.8 m), GECCO2 (0.9 m), GloSea5 (0.9 m), MOVE-CORE (0.9 m), TP4 (0.9 m), ECCO-v4 (1.0 m), ECDA (1.0 m), MERRA-Ocean (1.0 m), G2V3 (1.1 m), MOVE-G2 (1.1 m) and UR025-4 (1.1 m).

For a detailed overview on how each reanalysis is linked to each observational dataset, in terms of RMSE, MRSS and R, the
reader is referred to the tables presented in Appendix B.

## 3.2  Comparison of reanalyses to each other

As a first assessment of how well the reanalyses compare to each other, we estimate the RMSE and R between time series of SIT anomaly, at every grid point and between all pairs of products. The results are organized as a square matrix in Fig. 5, where the number on the top of each panel represents the respective global value estimated by considering the data from all grid
points. The lower triangular part of the matrix reveals that the smallest RMSE is found for the pair ECDA–UR025-4 (RMSE = 0.21 m). Only four other pairs present RMSE ≤ 0.25, they are the match between the two GloSea5 products (0.23 m), and the combination of UR025-4 against C-GLORS05 and ECCO-v4 (0.25 m). The largest error takes place when comparing GECCO2–MOVE-G2 (0.61 m).

From Fig. 5 (lower triangle), the averaged RMSE for each individual reanalyses indicates that UR025-4 is the reanalysis
closer to the ensemble, while MOVE-G2 has the largest errors compared to its counterparts: UR025-4 (0.30±0.06 m), ECCO-v4 (0.33±0.06 m), ECDA (0.33±0.06 m), GloSea5 (0.34±0.07 m), C-GLORS05 (0.35±0.06 m), PIOMAS (0.35±0.06 m), MOVE-CORE (0.36±0.06 m), GloSea5-GO5 (0.36±0.07 m), TP4 (0.37±0.06 m), ORAP5 (0.38±0.06 m), G2V3 (0.41±0.06 m), MERRA-Ocean (0.44±0.04 m), GECCO2 (0.45±0.06 m), MOVE-G2 (0.47±0.06 m).





**Figure 4.** Comparison between sea ice thickness from reanalyses and sea ice thickness (green points) or draft (black points) from observational datasets. The lines represent the linear fits having the reanalysis as the predictor and the observations as predicted variables. The Mean Residual Sum of Squares (MRSS) from the fit, the correlation coefficient (R) and the Root Mean Squared Error (RMSE) are also displayed for each comparison. RMSE is calculated only when comparing SIT from both sources (green), but not when comparing SIT and draft (black). Reanalyses labeled in blue and red highlight whether the datasets were built with or without sea ice data assimilation, respectively.

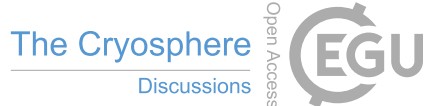





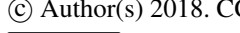



**Figure 5.** Square matrix-plot displaying the Root Mean Squared Error (RMSE) (lower triangular part) and the Correlation Coefficient (R) maps (upper triangular part), estimated from the sea ice thickness time series, at every grid point and between all pairs of reanalyses. The numbers on the top of each panel indicate the respective value calculated with data from all grid points. All maps have the 0°-longitude placed at 6-o'clock, while the bounding latitude is 67°N. Reanalyses labeled in blue and red highlight whether the datasets were built with or without ice data assimilation, respectively.

At the regional scale, most of the pairs of reanalyses have larger differences off the northern Greenland coast and to the north of the Canadian Archipelago, which are more pronounced in the MERRA-Ocean product. Almost all systems present minimum errors in the central Arctic Basin.

In turn, the upper triangular part of the matrix in Fig. 5 displays the linear relationship between pairs of reanalyses, quantified by the correlation coefficient. The strongest pan-Arctic correlations are observed for GloSea5–Glosea-G05 (R = 0.69), ORAP5–UR025-4 (0.67) and G2V3–ORAP5 (0.65). MOVE-CORE and MOVE-G2 present a marked anti-correlation with several other reanalyses, mainly in the central Arctic Ocean. Such anti-correlation is also reflected in the sea ice volume anomalies shown in Fig. 1. Notice that negative anomalies in MOVE-CORE and MOVE-G2, for instance from 2001 to 2004, are associated with strong positive anomalies in reanalyses as GECCO2, G2V3 and ECDA, as well as ORAP5, PIOMAS, TP4 and UR025-4, though in a less pronounced way (Figs. 1 and 5). We do not have a clear understanding of why these anti-correlations take place.

## 3.3 Time scales

An important property inherent to time series in general concerns their time scale and/or persistence, as defined in Sec. 2.3. In other words, we aim to infer for how long the SIT anomaly maintains a good correlation with future measurements at the same grid cell. Persistence can also be perceived as part of a self-prediction scheme where past data is used to predict future values. In addition, it is a relevant variable to be taken into account when designing the sampling frequency of observational programmes, specially if these programmes target the understanding of the SIT variability.

Fig. 6 displays the $e$-folding time scales for the SIT anomaly at every grid point, and for all reanalyses. The Area Weighted Mean (AWM) time scales (in months) sorted in ascending order are: 2.5 (GloSea5), 2.6 (GloSea5-GO5), 3.6 (PIOMAS) 3.7 (ECCO-v4), 3.8 (MERRA-Ocean), 4.0 (UR025-4), 4.3 (TP4), 4.4 (C-GLORS05), 4.7 (ORAP5), 4.9 (MOVE-CORE), 5.0 (G2V3), 6.0 (ECDA), 7.2 (MOVE-G2), and 7.8 months (GECCO2). These values were calculated taking into account only grid points with a valid SIT value from all reanalyses.

The results reveal that the thickness anomalies from reanalyses with no ice data assimilation (NA; Fig. 6, red labels) present a longer persistence, mainly distinguished in MOVE-G2 and GECCO2. Potential reasons to explain why the thickness anomalies persist longer in NA systems are suggested and discussed in Sec. 4. On the contrary, the thickness anomalies from the GloSea5 systems (GloSea5 and GloSea5-GO5) have a much shorter persistence.

From a regional point of view, Fig. 6 shows that GloSea5 and GloSea5-G05 are the only reanalyses in which the SIT anomaly persistence is remarkably short all over the Arctic, presenting $e$-folding time scales higher than 4 months only in a few, not





evenly distributed, grid points. By contrast, the SIT from GECCO2 has a marked longer persistence (>15 months) extending from the region off the northen coast of Greenland to the north of the Canadian Archipelago and mid Arctic Ocean. The ECDA product presents a relatively similar pattern of time scale over the region mentioned above, but persisting for a shorter period (∼8 months). SIT anomalies from MOVE-G2 also indicate long persistence off the northern Greenland coast, extending to the

central Arctic and East Siberian Sea. The time scale maps from G2V3, ORAP5, PIOMAS, TP4 and UR025-4 present a slight resemblance, but not so clear as between GloSea5 and GloSea5-G05.

Nevertheless, the results above should be interpreted with caution. The $e$-folding time scale is a metric that depends on the shape of the lagged-autocorrelation curve, which in turn may differ according to the period and time span of the original time series being analyzed. In order to evaluate how stable the time scale is by varying the time span and also by evolving

over time, we applied a time-moving and length-variable window to calculate the $e$-folding time scale of the ice volume anomaly (detrended in the same way as the SIT time series) from the two longest reanalyses (GECCO2 and MOVE-CORE), as shown in Figs. 7a,e. The window length varies from 5 to 59 years (stepped by 1 year) and it moves over time stepped forward by one month. Here, we use the ice volume anomaly, rather than SIT anomaly, for two reasons. First, because it is computationally more efficient than calculating the time scale for the SIT anomalies at every grid cell, considering the large

number of interactions for a time-moving and length-variable window. Second, because the volume provides a pan-Arctic perspective of the SIT persistence.

Notice in Figs. 7a,e that the persistence overall grows to longer than ∼20 years when taking into account long time spans, remarkably for GECCO2 in which ice volume anomaly persists for longer than 25 years at several center times. As for the thickness anomalies, MOVE-CORE presents a shorter persistence compared to GECCO2.

As a measure of stability, we estimate the standard deviations for all computations displayed in Figs. 7a,e. Results show that MOVE-CORE has a more stable time scale, with standard deviation of 3.0 months from its mean (9.7 months), while GECCO2 presents average and standard deviation of 15.0±6.5 months.

Figs. 7b,f show the case where the window length is 15 years, as it is for the overlapping period Jan/1993–Dec/2007. For this case, the average (standard deviation) time scales for GECCO2 and MOVE-CORE are 11.4±2.6 and 9.1±2.5 months,

respectively. Minimum to maximum ranges are 6.2–16.5 months for GECCO2 and 4.9–13.5 months for MOVE-CORE. If we take into account the same center time of the time span Jan/1993–Dec/2007, that is mid-Jun/2000, the ice volume anomaly persistences are 13.6 and 9.2 months (red stars in Figs. 7b,f, respectively). Note that the time scales of the ice volume anomalies are a few months longer compared to the persistence of the AWM-thickness anomalies (9.2 months, GECCO2; 5.4 months, MOVE-CORE).

We make use of wavelet analysis (Torrence and Compo, 1998) to evaluate whether the time series displayed in Figs. 7b,f exhibit significant band(s) of variability. Figs. 7c,d reveal that the ice volume anomaly from GECCO2 presents two bands of significant variability, as highlighted by the horizontal gray bars in Fig. 7d. The first spans from 4.4 to 6.1 years, and it is present in the first half of the time series but does not persist over time (black contours in Fig. 7c). The second is marked by periods longer than 10.7 years, which seems to be recurrent over time, but should be interpreted with caution since it is placed

near the "cone of influence", where edge effects become important, as indicated by cross-hatched areas overlapping the black



**Figure 6.** E-folding time scales (or persistence) estimated for the SIT time series. Only grid cells in which the time-mean (for the Jan/1993–Dec/2007 period) SIT at the time of summer minimum is greater than 0.1 m are taken into account for the computations. Averages for the systems with ice Data Assimilation and No data Assimilation are represented by the NA and DA panels. All maps have the 0°-longitude placed at 6-o'clock, while the bounding latitude is 67°N. Reanalyses labeled in blue and red highlight whether the datasets were built with or without ice data assimilation, respectively.



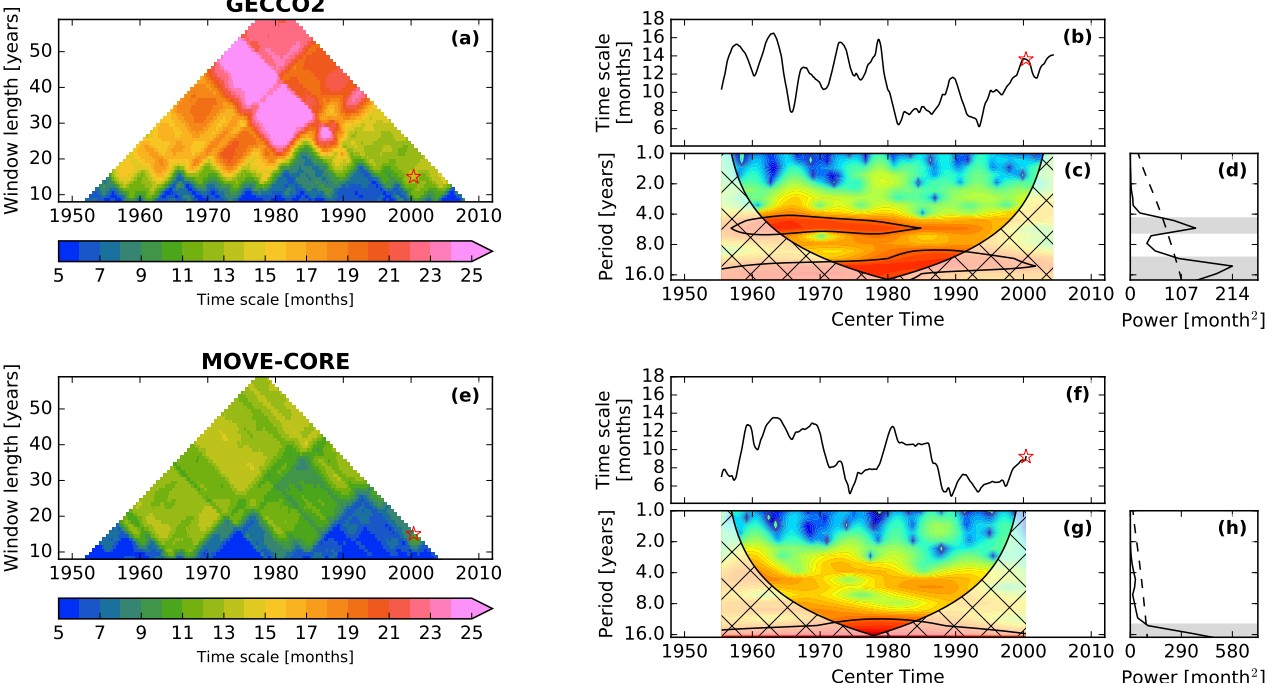

**Figure 7.** (a) Moving $e$-folding time-scales estimated for the ice volume anomaly time series from the GECCO reanalysis. The window length varies from 5 to 59 years and it is stepped forward by one month over a total period of 64 years (Jan/1948–Dec/2011). (b) Moving $e$-folding time-scales for the 15-year window length case. The red stars in (a) and (b) indicate the 15-year overlapping period (Jan/1993–Dec/2007, center time mid Jun/2000). (c) Wavelet power spectrum of (b), with Morlet as wavelet mother. The black lines denote the 95% significance levels above a red noise background spectrum, while the cross-hatched areas indicate the "cone of influence" where edge effects become important. The colorbar is omitted in panel (c) since we are not interested in the power's magnitude but in the frequencies outstanding as significant in the spectrum. (d) Time-integrated power spectrum from the wavelet analysis, where the dashed line corresponds to the 95% significance level. The bands of significant periods (4.4–6.1 years and >10.7 years) are highlighted by the gray horizontal bars. (e–h) Same as (a–d), respectively, but for the MOVE-CORE ice volume anomaly which has a spanning period of 60 years (Jan/1948–Dec/2007). The horizontal gray bar in (h) highlights the only period band of significant variability, being defined by periods longer than 12.7 years.

contours in Fig. 7c. The ice volume anomaly from MOVE-CORE, in turn, is marked by a single band of significant variability, with periods longer than 12.7 years (Figs. 7g,h). Again, this band should be interpreted with caution since it is also placed near the "cone of influence".

## 3.4 Length scales

5   The $e$-folding length scale is a metric used for indicating how well a variable from a certain grid cell compares to the neighboring cells. As for the time scale, the length scale is a promising parameter to be explored when designing observational





systems, but in terms of spatial coverage of instruments. Simplistically, regions marked with high length scales would require less instruments to be better monitored.

Fig. 8 shows the length scales for the SIT anomaly at every grid point. The AWM-length scales, in kilometers (km) and ascending order, for each system are: 337.0 (GloSea5), 420.7 (GloSea5-GO5), 544.6 (C-GLORS05), 681.5 (MERRA-Ocean),
724.3 (TP4), 728.2 (G2V3), 596.9 (ORAP5), 597.4 (UR025-4), 730.2 (PIOMAS), 732.5 (ECCO-v4), 846.7 (MOVE-G2), 835.8 (MOVE-CORE), 934.0 (ECDA), 935.7 km (GECCO2).

A similar pattern to the time scale is here observed, with GloSea5 and GloSea5-GO5 presenting the minimum length scales, rarely higher than 500 km, while the reanalyses without sea ice data assimilation are characterized by higher length scales, sometimes higher than 1200 km. In all systems the length scales are relatively longer near the central Arctic. This suggests
that higher length scales are associated with thicker ice. The relationships between mean ice thickness *vs.* time scale *vs.* length scale will be explored in details in Sec. 4.

The stability of the length scale over time (Fig. 9) was tested by means of a moving window with 15 years length, as follows: first, we calculate the one-point correlation maps for every grid point; second, we estimate the length scale for each one-point correlation map; third, the AWM-length scale was calculated taking into account only grid points with a valid SIT value
from all reanalyses; fourth, the process was repeated by stepping forward the 15-year window by 12 months. It is worthwhile mentioning that, computationally, it is much more expensive to calculate the length scale than the time scale. This is the reason why, here, we just use a time-moving but not length-variable window. The results suggest that the length scale is relatively more stable than the time scale (Fig. 7b,f), for what a quantification is presented in the conclusions of this study.

## 4 Discussion and conclusions

The first aim of this study was to evaluate how the SIT from the reanalyses compares against observational datasets, either draft or SIT. We have used three different metrics to perform this comparison: the correlation coefficient (R), as a measure of the linear correlation between datasets; the Mean Residual Sum of Squares (MRSS), as an indicator of whether reanalysis values are good predictors for the observations; and the Root Mean Square Error (RMSE), which directly compares how the SIT from the reanalyses approaches the SIT from observations. The results show that some of the reanalyses have a relatively
good correspondence either comparing SIT and draft or SIT from both sources of data. This is the case, for instance, for the TP4 product. A direct comparison between SIT from both sources indicates RMSEs ranging from 0.7 to 1.1 m. PIOMAS has the best agreement with observations. An interesting case is GECCO2, which presents a relatively small RMSE, as well as good correlation and linear relationship to the SIT observational datasets. However, this same product is weakly correlated and has poor predictive skill to the draft observational datasets.

One of our main goals in performing such a comparison was to identify whether or not systems built with assimilation of sea ice concentration data are closer to observations, compared to the products built with no sea ice data assimilation. The results are not conclusive and reanalyses with sea ice data assimilation do not necessarily perform better. One could speculate that some reanalyses do not take the best advantage of the covariances between sea ice concentration and SIT.



**Figure 8.** E-folding length scales estimated for the SIT time series. Only grid cells in which the time-mean (for the Jan/1993–Dec/2007 period) SIT at the time of summer minimum is greater than 0.1 m are taken into account for the computations. Averages for the systems with ice Data Assimilation and no Data Assimilation are represented by the NA and DA panels. All maps have the 0°-longitude placed at 6-o'clock, while the bounding latitude is 67°N. Reanalyses labeled in blue and red highlight whether the datasets were built with or without ice data assimilation, respectively.



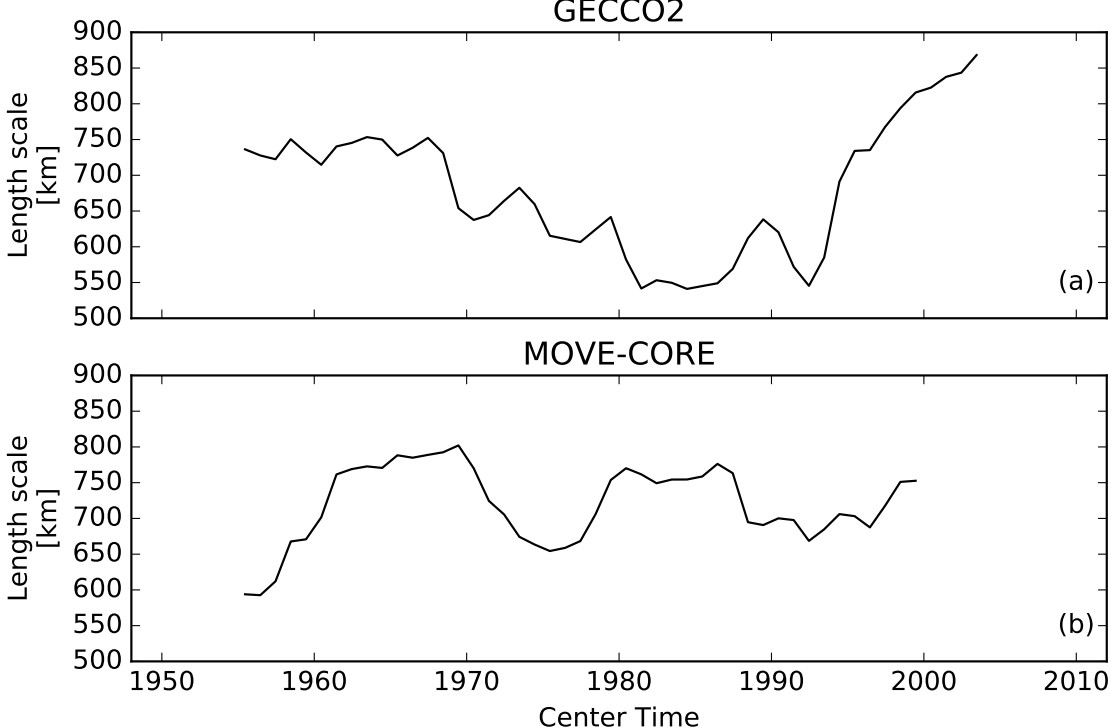

**Figure 9.** (a) Moving *e*-folding length scales estimated for the ice volume anomaly time series from the GECCO reanalysis. The window length is 15 years and it is stepped forward by 12 months over a total period of 64 years. (b) Same as (a), but for the MOVE-CORE reanalysis, which has a time span of 60 years.

It is worthwhile mentioning that the comparison among SIT from the different reanalyses is not straightforward and does not necessarily improve due to common specifications and key parameters from the two systems being compared. For instance, the pair C-GLORS05–G2V3 presents relatively high RMSE (0.39 m), and not too strong correlation (R = 0.4), even though both systems share a set of specifications and parameters such as ocean–sea ice model, atmospheric forcing, vertical discretization, 5 number of ice thickness categories, dynamics-EVP, ocean-ice drag coefficient, analysis window and also both assimilate sea ice data. The same is valid for other pairs of systems, e.g., G2V3–ORAP5.

The pair with smallest RMSE, ECDA–UR025-4, has at the same time a relatively weak linear relationship (R = 0.21). This reinforces the importance of looking at different metrics, when comparing different products. If we average the RMSEs that one specific reanalysis present against all the others, and so compare with the pan-Arctic mean ice volume of this same reanalyses, 10 it becomes clear that products with relatively low sea ice volume (i.e., thin ice) present small RMSEs when compared with their counterparts (Fig. 10). This helps to explain why ECDA and UR025-4 have a small RMSE (Fig. 10), though their anomalies are marked by a not so strong correlation. This is also evident in the respective sea ice volume anomalies from these both




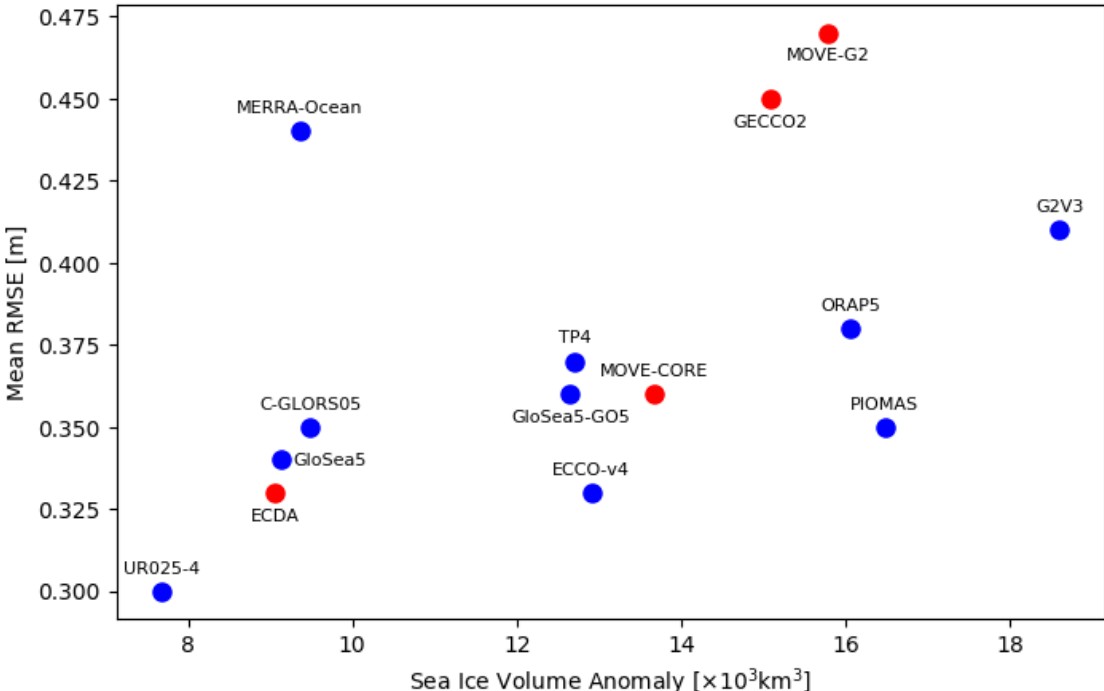

**Figure 10.** Time-mean sea ice volume *versus* the mean RMSE. This last parameter is an average of the RMSEs that each reanalysis has when compared individually to the other thirteen reanalyses. Reanalyses in blue and red highlight whether the datasets were built with or without ice data assimilation, respectively.

reanalyses shown in Fig. 1. MERRA-OCEAN is an outlier in this pattern, presenting a thin sea ice but large RMSE compared to the other reanalyses.

Another main goal of this work was to characterize the time and length scales of the sea ice thickness, as well as to report whether these parameters are influenced by the fact that a respective reanalysis assimilates or not sea ice concentration data. In this case, sea ice data assimilation plays a clear role in the referred scales: systems with sea ice data assimilation are characterized by shorter time and length scales compared to the systems which do not assimilate sea ice data. Likely, the main reason why this happen is linked to the fact that when a reanalysis assimilates sea ice information, the system is forced towards the assimilated conditions, differently from what occurs with free-running models. Eventually, data assimilation introduces SIT increments that are not necessarily physical, and so contributes to an attenuation in the correlation of this variable at a certain grid cell both in time, with their future estimations, and in space, with the neighboring grid points. Fig. 11 shows an interesting example where the time scales are estimated for the same system, but built with (G2V3) and without (G2V1) assimilation of sea ice concentration. Again, the results show relatively longer time scales for the system with no data assimilation.

The time scales have a marked anti-correlation with the sea ice velocities as shown in Fig. 12: reanalyses with faster sea ice present a short time scale and the other way around. The same is observed for the length scales, though not so marked as





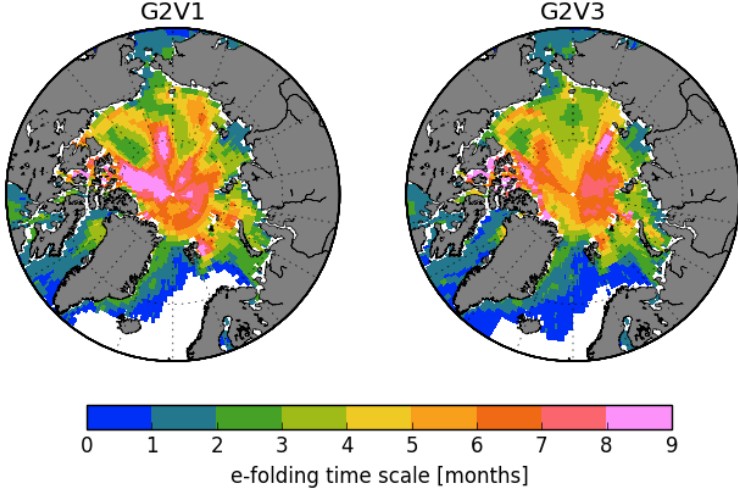

**Figure 11.** Time scale estimated to the same reanalysis, but built with (G2V3) and (G2V1) without data assimilation.

for time scales (not shown). Different specifications from the reanalyses influence the sea ice velocity. High air–ice and low ocean–ice drag coefficients contribute to faster ice velocities, respectively (Tandon et al., 2018). For instance, ECCO-v4 has the second highest air–ice (smaller only compared to MOVE-CORE) and the smallest ocean–ice drag coefficients (see Table 1 from Chevallier et al. (2017)). This may explain why ECCO-v4 has relatively high ice velocities (Fig. 12) and, therefore,

low time and length scales (Figs. 6 and 8). In addition, the ice strength formulation is a major player in the sea ice velocity (Ungermann et al., 2017). All reanalyses follow the linear parameterization proposed by Hibler (1979), except the GloSea5 products which employ the ice strength formulation following Rothrock (1975). Ungermann et al. (2017) presented a detailed study comparing both methods and have shown that, for systems characterized by relatively thinner ice, model simulations with Rothrock (1975) formulation result in lower ice strength, and therefore faster ice velocities, compared to Hibler (1979)

formulation. The combination between relatively thin ice (see colorbar in Fig. 13) and Rothrock (1975) ice strength formulation is a potential explanation of why GloSea5 and GloSea5-GO5 have such a short time and length scales.

We have shown that time and length scales are influenced by whether or not the reanalyses assimilate sea ice data. However, are these two properties correlated to each other? Also, are they somehow related to the mean SIT of the system? Fig. 13 shows that long time scales are associated with large length scales. This figure also compares the respective scales against the

time-mean ice volume. Interestingly, for the reanalyses with ice data assimilation, an increase in mean ice thickness seems to lead to an increase in time and length scales, as it is suggested by the points in the bottom-left quadrant defined by the gray dashed lines (Fig. 13). This is in agreement with the higher time and length scales associated with slower ice shown in Fig. 12, suggesting that by increasing ice thickness, ice moves slower, resulting in higher length and time scales. The extreme example is G2V3 which has the thickest ice and also relatively high length and time scales. This does not seem to be the case for the

systems which do not assimilate ice data as indicated by the points in the upper-right quadrant.




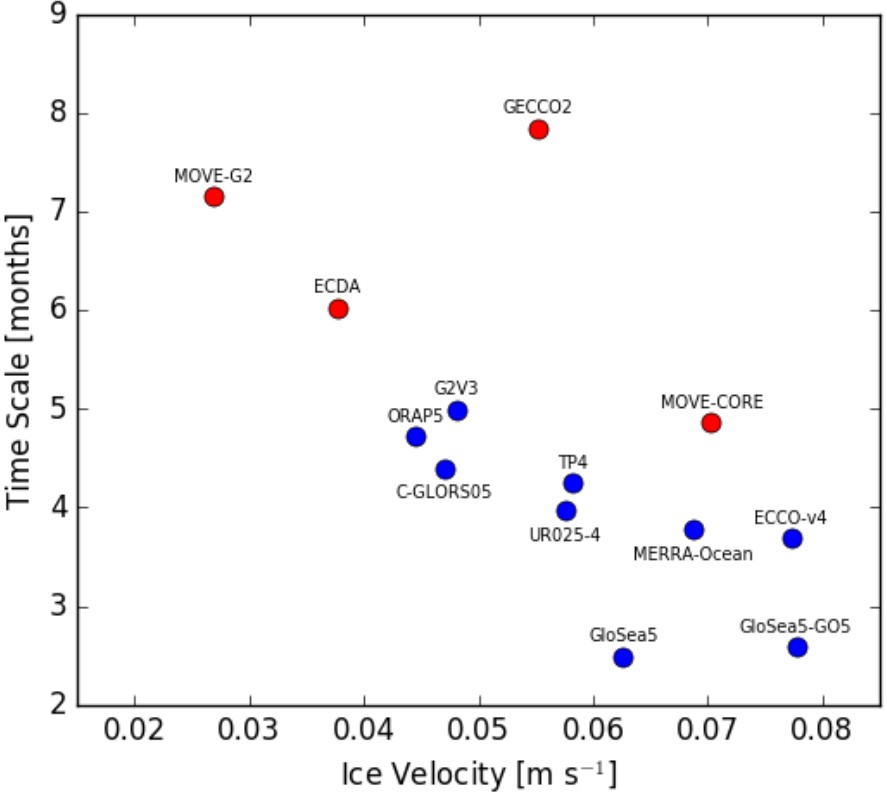

**Figure 12.** Scatter plot of the Average Weighted Mean (AWM) sea ice velocity *versus* the AWM time scale. Reanalyses in blue and red highlight whether the datasets were built with or without ice data assimilation, respectively.

As mentioned before, the ice thickness time and length scales are interesting properties to be explored when designing and planing an optimal observation system both in terms of temporal sampling and spatial placement of instruments. Nevertheless, Sec. 3.3 (Fig. 9) and Sec. 3.4 (Fig. 7) showed that these properties also vary over time. The over-time stability of time and length scales can be quantified by the coefficient of variation ($C_v$=std/mean, where std is the standard deviation). The $C_v$ is a non-dimensional metric used to evaluate the extent of a certain variability in relation to its mean, allowing to compare different properties. The $C_v$ for the GECCO2 and MOVE-CORE moving time scales, estimated from the time series shown in Fig. 7b,f, are 0.23 and 0.27, respectively, while the $C_v$ for moving length scales (Fig. 9) are 0.13 and 0.07.

Lastly, it is worthwhile mentioning that we still consider both time and length scales promising properties to support the design of an optimal observing system. As suggested by the $C_v$ presented above, the length scale is considerably more stable than the time scale so that it is a more reliable variable to be taken into account. For instance, the multiple linear regression model used by Lindsay and Zhang (2006), for determining optimal locations to predict sea ice extent from SIT, could be combined with the length scale information, avoiding that two or more stations placed into the same radius of correlation



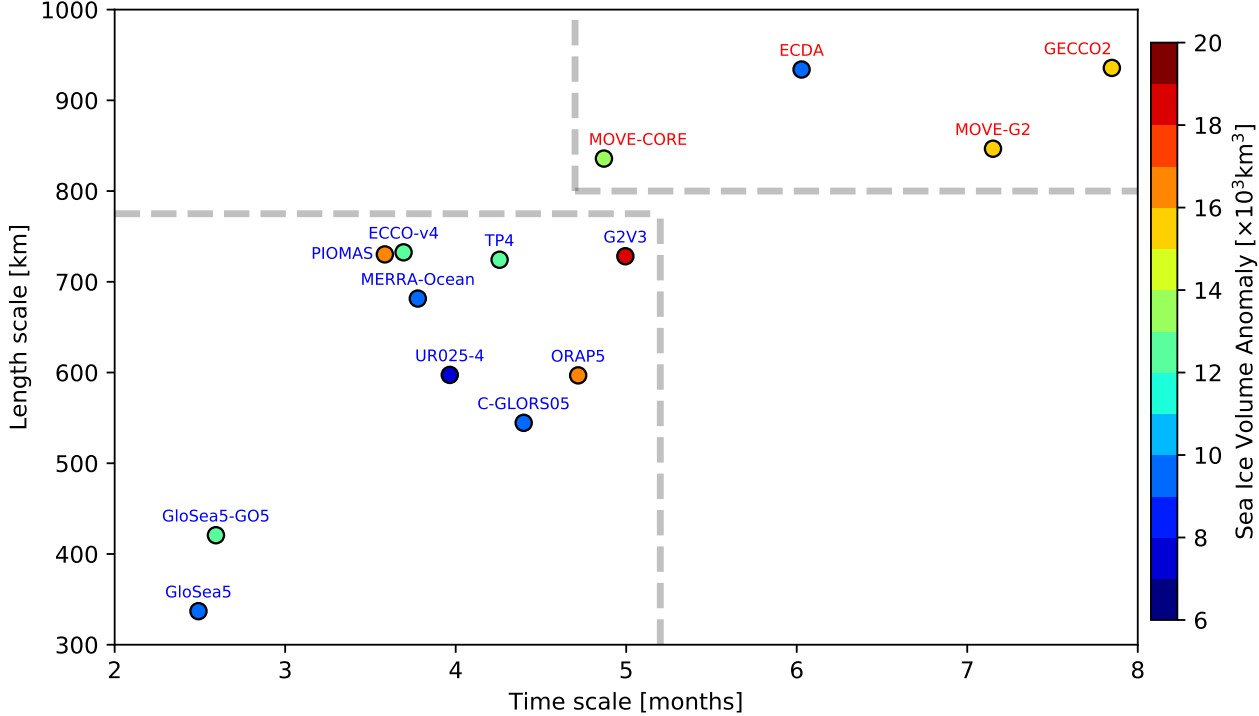

**Figure 13.** Scatter plot of the Average Weighted Mean (AWM) time scale *versus* the AWM length scale, with colors representing the time-mean sea ice volume. Reanalyses labeled in blue and red highlight whether the datasets were built with or without ice data assimilation, respectively. The dashed gray lines have just an illustrative purpose, in order to highlight the longer time scale and larger length scale for systems without data assimilation.

(length scale) are selected. Time scale would be more useful if used in combination with the knowledge of its variability. Further specific studies are required to evaluate the performance of time and length scales in providing support for the optimal design of observational programmes, though this work already shows some promising results in such direction.





## Appendix A: List of acronyms

This appendix displays all acronyms and their respective long names referred to in the text and used in the figures. The long names of some acronyms were previously omitted in order to preserve the readability of text, while others were already defined. All of them will be mentioned below so that the reader can easily consult their meaning at any time. As follows:

AER        Atmospheric and Environmental Research

     AWI        Alfred Wegener Institute

     AWM        Area-Weighted Mean

     BIO-LS        Bedford Institute of Oceanography Lancaster Sound

     BGEP        Beaufort Gyre Exploration Project

C-GLORS05        CMCC - Global Ocean Reanalysis System

     CDR        Climate Data Record

     CHK        Chuck Sea

     CMCC        Centro Euro-Mediterraneo sui Cambiamenti Climatici

     CryoSat        CRYOgenic SATellite

$C_v$        Coefficient of variation

     $DA_v$        Data Assimilation

     EBS        Eastern Beaufort Sea

     ECCO-v4        Estimating the Circulation and Climate of the Ocean - version 4

     ECDA        Ensemble Coupled Data Assimilation

ECMWF        European Centre for Medium-Range Weather Forecasts

     ESA        European Space Agency

     EVP        Elastic-Viscous-Plastic

     GECCO2        German - Estimating the Circulation and Climate of the Ocean

     GFDL        Geophysical Fluid Dynamics Laboratory

GLORYS        Global Ocean reanalysis and Simulation





| | | |
|---|---|---|
| | GloSea5 | Global Seasonal forecasting system |
| | GloSea5-GO5 | Global Seasonal forecasting system - Global Ocean 5.0 |
| | GMAO | Global Modeling and Assimilation Office |
| | GS | Greenland Sea |
| 5 | GSFC | Goddard Space Flight Center |
| | G2V3 | GLORYS 2 - version 3 |
| | ICESat | Ice, Cloud,and land Elevation Satellite |
| | IOS | Institute of Ocean Science |
| | JMA | Japan Meteorological Agency |
| 10 | JPL | Jet Propulsion Laboratory |
| | MERRA | Modern Era Retrospective-Analysis for Research and Applications |
| | ARC MFC | Arctic Marine Forecasting Center |
| | MIT | Massachusetts Institute of Technology |
| | MOVE-CORE | Multivariate Ocean Variational Estimation - Coordinated Ocean-ice Reference Experiment |
| 15 | MOVE-G2 | Multivariate Ocean Variational Estimation - Global version 2 |
| | MRI | Meteorological Research Institute |
| | MRSS | Mean Residual Sum of Squares |
| | $NA_v$ | No data Assimilation |
| | NASA | National Aeronautics and Space Administration |
| 20 | NOAA | National Oceanic and Atmospheric Administration |
| | NPEO | North Pole Environmental Observatory |
| | ORA-IP | Ocean Reanalysis Intercomparison project |
| | PIOMAS | Pan-Arctic Ice-Ocean Modeling and Assimilation System |
| | R | Correlation Coefficient |



| | | |
|---|---|---|
| | RMSE | Root Mean Squared Error |
| | SIT | Sea Ice Thickness |
| | TP4 | TOPAZ4 |
| | UK-Subs-AN | UK Navy Submarines - Analog |
| 5 | UK-Subs-DG | UK Navy Submarines - Digital |
| | ULS | Upward-Looking Sonar |
| | US-Subs-AN | US Navy Submarines - Analog |
| | US-Subs-DG | US Navy Submarines - Digital |
| | UR025-4 | University of Reading, 1/4° deg - version 4 |
| 10 | VP | Viscous-Plastic |
| | YOPP | Year Of Polar Prediction |





**Appendix B: Tables**

This appendix presents the comparison of all reanalysis products with all different observational datasets. Such comparison is based on three different metrics: Root Mean Square Error (RMSE), Mean Residual Sum of Squares (MRSS) and Correlation Coefficient (R).





**Table B1.** Root Mean Square Error (RMSE; in m) calculated between the SIT from observations and reanalyses. The RMSE was not calculated for the draft datasets (*).

| (Reanalyses) | NPEO* | BGEP* | IOS-EBS* | IOS-CHK* | SSUB-AN* | USSUB-DG* | UKSUB-AN* | UKSUB-DG* | AWI-GS* | BIO-LS* | Davis-St* | CanCoast* | IceBridge-V2 | IceBridge-QL | ICESat1-G | CryoSat-AWI |
|---|---|---|---|---|---|---|---|---|---|---|---|---|---|---|---|---|
| C-GLORS05 | – | – | – | – | – | – | – | – | – | – | – | – | 0.7 (390) | – | 0.9 (28413) | 0.6 (29058) |
| ECCO-V4 | – | – | – | – | – | – | – | – | – | – | – | – | 0.9 (264) | – | 1.1 (29147) | 0.6 (5934) |
| ECDA | – | – | – | – | – | – | – | – | – | – | – | – | 1.6 (926) | 1.3 (439) | 1.1 (29102) | 1.0 (76909) |
| G2V3 | – | – | – | – | – | – | – | – | – | – | – | – | 1.0 (390) | – | 1.5 (28413) | 0.7 (29058) |
| GECCO2 | – | – | – | – | – | – | – | – | – | – | – | – | 1.2 (399) | – | 0.9 (27560) | 0.8 (30891) |
| GloSea5 | – | – | – | – | – | – | – | – | – | – | – | – | 1.5 (670) | 1.3 (201) | 1.1 (29071) | 0.8 (53005) |
| GloSea5-GO5 | – | – | – | – | – | – | – | – | – | – | – | – | 1.0 (910) | 1.0 (671) | 1.0 (29071) | 0.8 (101407) |
| MERRA-Ocean | – | – | – | – | – | – | – | – | – | – | – | – | 1.2 (666) | 0.9 (202) | 1.1 (28937) | 1.0 (45491) |
| MOVE-CORE | – | – | – | – | – | – | – | – | – | – | – | – | – | – | 0.9 (24966) | – |
| MOVE-G2 | – | – | – | – | – | – | – | – | – | – | – | – | 1.4 (651) | 1.4 (199) | 1.1 (28413) | 1.0 (52074) |
| ORAP5 | – | – | – | – | – | – | – | – | – | – | – | – | 0.9 (683) | 0.9 (204) | 0.9 (29366) | 0.7 (53293) |
| PIOMAS | – | – | – | – | – | – | – | – | – | – | – | – | 1.1 (933) | 0.8 (860) | 0.9 (29452) | 0.6 (127270) |
| TP4 | – | – | – | – | – | – | – | – | – | – | – | – | 1.2 (914) | 0.9 (442) | 1.0 (28882) | 0.8 (76046) |
| UR025-4 | – | – | – | – | – | – | – | – | – | – | – | – | 1.5 (258) | – | 1.1 (29071) | 0.7 (5922) |





**Table B2.** Mean Residual Sum of Squares (MRSS; in m$^2$) estimated from the linear fit between reanalysis SIT (predictor) and observations (predict), either draft or SIT, parameters, respectively. Draft observational datasets are distinguished from the SIT datasets by the (*).

| Reanalyses | NPEO* | BGEP* | IOS-EBS* | IOS-CHK* | SSUB-AN* | USSUB-DG* | UKSUB-AN* | UKSUB-DG* | AWI-GS* | BIO-LS* | Davis-St* | CanCoast* | IceBridge-V2 | IceBridge-QL | ICESat1-G | CryoSat-AWI |
|---|---|---|---|---|---|---|---|---|---|---|---|---|---|---|---|---|
| C-GLORS05 | 0.3 (64) | 0.1 (309) | 0.5 (340) | 0.1 (26) | 0.1 (216) | 0.3 (766) | 1.5 (40) | – | 0.3 (112) | – | 0.2 (54) | 0.2 (14) | 0.4 (390) | – | 0.7 (28413) | 0.2 (29058) |
| ECCO-V4 | 0.4 (64) | 0.1 (265) | 0.4 (340) | 0.1 (26) | 0.2 (216) | 0.4 (701) | – | – | 0.2 (110) | 0.4 (36) | 0.2 (48) | 0.2 (195) | 0.5 (264) | – | 0.8 (29147) | 0.2 (5934) |
| ECDA | 0.4 (64) | 0.1 (369) | 0.5 (382) | 0.1 (26) | 0.4 (794) | 0.4 (1000) | 2.3 (149) | 0.1 (27) | 0.3 (131) | 0.4 (36) | 0.3 (53) | 0.2 (2664) | 0.9 (926) | 0.7 (439) | 0.8 (29102) | 0.4 (76909) |
| G2V3 | 0.6 (64) | 0.2 (310) | 0.4 (324) | 0.2 (26) | 0.7 (216) | 0.4 (637) | – | – | 0.2 (108) | – | 0.2 (54) | – | 0.5 (390) | – | 0.6 (28413) | 0.3 (29058) |
| GECCO2 | 0.6 (64) | 0.2 (308) | 0.6 (357) | 0.2 (23) | 0.9 (840) | 0.8 (100) | 2.0 (149) | 0.2 (27) | 0.3 (115) | 0.4 (36) | 0.4 (46) | 0.3 (2864) | 0.5 (399) | – | 0.6 (27560) | 0.2 (30891) |
| GloSea5 | 0.6 (64) | 0.2 (348) | 0.4 (382) | 0.1 (26) | 0.5 (253) | 0.4 (816) | 0.9 (40) | – | 0.3 (134) | – | 0.1 (67) | 0.2 (399) | 0.7 (670) | 0.6 (201) | 0.8 (29071) | 0.4 (53005) |
| GloSea5-GO5 | 0.5 (64) | 0.2 (371) | 0.4 (382) | 0.1 (26) | 0.5 (253) | 0.4 (777) | 0.8 (40) | – | 0.3 (134) | – | 0.1 (67) | 0.2 (376) | 0.7 (910) | 0.8 (671) | 0.8 (29071) | 0.4 (101407) |
| MERRA-Ocean | 0.4 (64) | 0.3 (328) | 0.5 (379) | 0.3 (26) | 0.8 (540) | 0.8 (1001) | 1.9 (149) | – | 0.2 (133) | 0.3 (36) | 0.1 (66) | 0.3 (857) | 0.5 (666) | 0.4 (202) | 0.7 (28937) | 0.4 (45491) |
| MOVE-CORE | 0.1 (41) | 0.2 (167) | 0.4 (379) | 0.2 (26) | 0.4 (843) | 0.5 (1001) | 1.9 (149) | 0.2 (27) | 0.2 (129) | – | 0.2 (39) | 0.2 (412) | – | – | 0.7 (24966) | – |
| MOVE-G2 | 0.7 (64) | 0.4 (348) | 0.5 (342) | 0.1 (26) | 0.4 (216) | 0.4 (638) | – | – | 0.3 (100) | – | 0.2 (48) | – | 0.7 (651) | 0.5 (199) | 0.8 (28413) | 0.5 (52074) |
| ORAP5 | 0.3 (64) | 0.2 (340) | 0.4 (339) | 0.1 (26) | 0.8 (646) | 0.7 (1000) | 1.3 (149) | – | 0.2 (112) | 0.3 (35) | 0.2 (50) | 0.3 (1840) | 0.6 (683) | 0.5 (204) | 0.7 (29366) | 0.3 (53293) |
| PIOMAS | 0.5 (64) | 0.1 (371) | 0.4 (382) | 0.1 (26) | 0.4 (646) | 0.5 (1001) | 2.0 (149) | – | 0.3 (134) | 0.3 (36) | 0.2 (67) | 0.3 (2789) | 0.8 (933) | 0.6 (860) | 0.7 (29452) | 0.3 (127270) |
| TP4 | 0.2 (64) | 0.2 (371) | 0.4 (345) | 0.2 (26) | 0.2 (216) | 0.3 (638) | – | – | 0.3 (115) | 0.3 (36) | 0.1 (67) | 0.1 (121) | 0.7 (914) | 0.6 (442) | 0.6 (28882) | 0.3 (76046) |
| UR025-4 | 0.5 (64) | 0.1 (273) | 0.4 (345) | 0.1 (26) | 0.3 (216) | 0.3 (637) | – | – | 0.3 (115) | – | 0.1 (67) | 0.2 (241) | 0.4 (258) | – | 0.7 (29071) | 0.2 (5922) |





**Table B3.** Correlation coefficient (R) estimated between the SIT from reanalysis and observations, either draft or SIT. Draft observational datasets are distinguished from the SIT datasets by the (*).

| Reanalyses | NPEO* | BGEP* | IOS-EBS* | IOS-CHK* | SSUB-AN* | USSUB-DG* | UKSUB-AN* | UKSUB-DG* | AWI-GS* | BIO-LS* | Davis-St* | CanCoast* | IceBridge-V2 | IceBridge-QL | ICESat1-G | CryoSat-AWI |
|---|---|---|---|---|---|---|---|---|---|---|---|---|---|---|---|---|
| C-GLORS05 | 0.73 (64) | 0.83 (309) | 0.49 (340) | 0.95 (26) | 0.92 (216) | 0.74 (766) | 0.35 (40) | – | 0.41 (112) | – | 0.80 (54) | -0.01 (14) | 0.47 (390) | – | 0.49 (28413) | 0.73 (29058) |
| ECCO-V4 | 0.65 (64) | 0.83 (265) | 0.54 (340) | 0.96 (26) | 0.87 (216) | 0.65 (701) | – | – | 0.55 (110) | -0.47 (36) | 0.82 (48) | 0.28 (195) | 0.31 (264) | – | 0.33 (29147) | 0.68 (5934) |
| ECDA | 0.68 (64) | 0.86 (369) | 0.52 (382) | 0.90 (26) | 0.81 (794) | 0.73 (1000) | 0.46 (149) | -0.65 (27) | 0.51 (131) | 0.27 (36) | 0.65 (53) | 0.62 (2664) | 0.20 (926) | 0.43 (439) | 0.29 (29102) | 0.48 (76909) |
| G2V3 | 0.35 (64) | 0.73 (310) | 0.65 (324) | 0.89 (26) | 0.44 (216) | 0.56 (637) | – | – | 0.61 (108) | – | 0.80 (54) | – | 0.34 (390) | – | 0.52 (28413) | 0.62 (29058) |
| GECCO2 | 0.40 (64) | 0.69 (308) | 0.34 (357) | 0.79 (23) | 0.36 (840) | 0.39 (1001) | 0.56 (149) | -0.28 (27) | 0.46 (115) | -0.39 (36) | 0.48 (46) | 0.17 (2864) | 0.45 (399) | – | 0.58 (27560) | 0.73 (30891) |
| GloSea5 | 0.32 (64) | 0.75 (348) | 0.61 (382) | 0.95 (26) | 0.55 (253) | 0.59 (816) | 0.69 (40) | – | 0.56 (134) | – | 0.87 (67) | 0.27 (399) | 0.40 (670) | 0.14 (201) | 0.28 (29071) | 0.58 (53005) |
| GloSea5-GO5 | 0.54 (64) | 0.81 (371) | 0.63 (382) | 0.95 (26) | 0.63 (253) | 0.59 (777) | 0.70 (40) | – | 0.59 (134) | – | 0.88 (67) | 0.21 (376) | 0.53 (910) | 0.53 (671) | 0.30 (29071) | 0.58 (101407) |
| MERRA-Ocean | 0.63 (64) | 0.65 (328) | 0.56 (379) | 0.79 (26) | 0.52 (540) | 0.43 (1001) | 0.60 (149) | – | 0.65 (133) | 0.66 (36) | 0.86 (66) | 0.22 (857) | 0.56 (666) | 0.57 (202) | 0.50 (28937) | 0.54 (45491) |
| MOVE-CORE | 0.80 (41) | 0.75 (167) | 0.65 (379) | 0.83 (26) | 0.78 (843) | 0.73 (1001) | 0.61 (149) | 0.33 (27) | 0.65 (129) | – | 0.71 (39) | 0.52 (412) | – | – | 0.47 (24966) | – |
| MOVE-G2 | -0.16 (64) | 0.48 (348) | 0.52 (342) | 0.93 (26) | 0.68 (216) | 0.63 (638) | – | – | 0.29 (100) | – | 0.76 (48) | – | -0.27 (651) | -0.41 (199) | 0.16 (28413) | 0.38 (52074) |
| ORAP5 | 0.79 (64) | 0.81 (340) | 0.60 (339) | 0.90 (26) | 0.50 (646) | 0.52 (1000) | 0.75 (149) | – | 0.52 (112) | 0.64 (35) | 0.77 (50) | 0.33 (1840) | 0.49 (683) | 0.43 (204) | 0.49 (29366) | 0.63 (53293) |
| PIOMAS | 0.48 (64) | 0.87 (371) | 0.70 (382) | 0.94 (26) | 0.79 (646) | 0.71 (1001) | 0.58 (149) | – | 0.60 (134) | 0.57 (36) | 0.74 (67) | 0.11 (2789) | 0.29 (933) | 0.68 (860) | 0.48 (29452) | 0.71 (127270) |
| TP4 | 0.81 (64) | 0.81 (371) | 0.67 (345) | 0.82 (26) | 0.86 (216) | 0.66 (638) | – | – | 0.52 (115) | 0.53 (36) | 0.91 (67) | 0.50 (121) | 0.39 (914) | 0.54 (442) | 0.57 (28882) | 0.63 (76046) |
| UR025-4 | 0.58 (64) | 0.84 (273) | 0.62 (345) | 0.95 (26) | 0.82 (216) | 0.75 (637) | – | – | 0.61 (115) | – | 0.86 (67) | 0.41 (241) | 0.41 (258) | – | 0.40 (29071) | 0.74 (5922) |





*Competing interests.* No competing interests are present

*Acknowledgements.* The work presented in this paper has received funding from the European Union's Horizon 2020 Research and Innovation programme under grant agreement No. 727862: APPLICATE project (Advanced prediction in Polar regions and beyond). The python wavelet software is provided by Evgeniya Predybaylo based on Torrence and Compo (1998) and is available at

5   URL: http://atoc.colorado.edu/research/wavelets/



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
