# Peer review of "On the time and length scales of the Arctic sea ice thickness anomalies: a study based on fourteen reanalyses"

_The Cryosphere, 2018_

## Referee Comment (RC1) · Anonymous Referee #1 · 2 Oct 2018

The arcticle of "On the time and length scales of the Arctic sea ice thickness anomalies: a study based on fourteen reanalyses" aims at the Arctic sea ice thickness (SIT) which contains considerable uncertainty in the popular 14 reanalyses. They evaluate the reproduced SITs from the reanalyses, and then investigate the e-folding time and length scales of the SIT anomalies. Clearly, these topic and the consequent findings are helpful to deep understanding the SIT and the concerned variability.

1) Undoubtedly, one conclusion is "reanalyses built with sea ice data assimilation present shorter time and length scales" . However, all the reanalyses were only assimilation of sea ice concentration, and the inferred conclusion is not based on the

direct comparison of with and without assimilation in a same system frame. The more proofs based on the sea ice concentration will be helpful to increase the rationality on physics. So the counterpart analysis on sea ice concentration shown in Fig. 2 and Fig. 3 will be robust.

2) Section 3.2 illustrates the intercomprison of the reanalyses. The current main features shown by Fig. 5 is not meaningful enough: some reanalyses are very close. . . In this section, more other information about SIT and its anomaly need to be added.

Firstly, the ensemble mean SITs based on with and without assimilation will be useful (P 15 Line 10: "This suggests that higher length scales are associated with thicker ice"). Furthermore, it will be complementary of the previous knowns in Uotila et al. (2018) and Johnson et al. (2012).

Johnson, M., Proshutinsky A., Aksenov Y., Nguyen A. T., Lindsay R., Haas C., Zhang J., Diansky N., Kwok R., et al.: Evaluation of Arctic sea ice thickness simulated by Arctic Ocean Model Intercomparison Project models. J. Geophys. Res., 117(C8), C00D31, doi:10.1029/2011JC007257, 2012.

Secondly, the standard deviations of the two ensembled SIT anomalies were not shown before and would be interested to the reader to know the variabilities or the distinguishes about the SIT anomaly in the reanalyses and even considering with and without assimilation.

3) Figure 2 clearly shows the time scale has been extended from less than 3 months to around 4 months, which is convert with the main finding (P1, Line 9: . . . data assimilation present shorter time and length scales).

4) The previously compared studies show atmospheric forcing fields essentially drive the results of sea ice simulations (Gerdes and Köberle, 2007; Hunke and Holland, 2007). Can you add some comment or analysis about the impact of the forcing resolutions on the SIT time and length scales?

Gerdes, R., and C. Köberle: Comparison of Arctic sea ice thickness variability in IPCC Climate of the 20th century experiments and in ocean–sea ice hindcasts, J. Geophys. Res., 112, C04S13, doi:10.1029/2006JC003616, 2007.

Hunke, E., and M. Holland: Global atmospheric forcing data for Arctic ice-ocean modeling, J. Geophys. Res., 112, C04S14, doi:10.1029/2006JC003640, 2007.

5) The ice draft measurements from submarine have been identified an overall over-estimation of +0.29 m (Rothrock and Wensnahan (2007). This dataset also is used in this study. Can you add some comments about these kinds of bias corrections (also to other related observational data) applied here or not.

Technical issues: 1) As a basic index calculated by the SIT, it still is not clear how to deal with the conflicts of seawater and ice cover at each grid. For example in Fig. 2, when the observed sea ice is lager 0.1m, but the reanalyses are not all covered by sea ice. It is also not clear at P 15 Line 13 when to calculate the correlation between the two points: how to ensure the same lengths of the SIT time series.

2) P3, Line 20: "The original horizontal grids range from 0.25 to 1". It is not correct because the reanalysis of TP4 is regional product with the resolution of 12-16km (also see Xie et al. (2017)).

Xie, J., Bertino, L., Counillon, F., Lisæter, K., and Sakov, P.: Quality assessment of the TOPAZ4 reanalysis in the Arctic over the period 1991-2013, Ocean Sci., 13(1), 123-144, doi:10.5194/os-13-123-2017, 2017.

3) P3, Line 27:"... in details by Chevallier et al. (2017) (their Table 1) and Balmaseda et al. (2015)". It is not correct because they did not include the TP4 product as least so recommend of the reference: Xie et al. (2017) or Uotila et al. (2018).

4) P4, Line 27:" the linear relationship between both parameters given by the hydro-static equation". It is better if clear to state the used equation or give a reference used.

5) Figure 1 adds the grid lines or labels the year on each panel. It is more convenient

to march the statement. So P 11, Line 13 "for instance from 2001 to 2004" looks not suitable, and can be replaced by "for instance from 2002 to 2004".

6) P 15 Line 10: "This suggests that higher length scales are associated with thicker ice" looks not so precis. It more likes around the North pole.

7) P 18 Line3:" .. scales of the sea ice thickness" replaced by "...scales of the sea ice thickness anomaly."

8) Figure 11 adds a panel to show the difference so that details would be more clear.

---

## Referee Comment (RC2) · Anonymous Referee #2 · 18 Oct 2018

The paper provides a summary of the sea ice thickness anomalies found in the most recent ocean-ice analyses. It considers the impact of the assimilation of sea ice observations within some of these systems to spatial and temporal scales of the anomalies.

In my opinion the paper has lots of potential and is almost there, but at the moment it is missing some extra synthesis/analysis which would make it a really useful reference for the observation and modelling community. The impact of sea ice assimilation was considered (but did not state if any of the models assimilate anything other the concentration, one may nudge the thickness too?); it would be helpful to understand how the impact of other choices in the system may also influence the sea ice thickness anoma-

lies. The sea ice models may have very different methods for modelling thickness. Do they have ice thickness distributions or a single category, does this have an impact? Different atmospheric forcing sets (with different imposed sea ice cover) may influence the local energy balance and in turn affect the ice thickness. I think considering a few other key elements of these systems would make this paper very helpful in guiding the use of reanalyses and their future development (both in terms of the type of data that is used but also the set up of the assimilation systems)

Specific comments:

The paper provides some details of the reanalyses you have studies but I think it is missing a table summarising the reanlayses - some form of synthesis would be beneficial to the reader, the papers that do tabulate some of this (e.g. Chevallier et al) are not complete for this reanalysis set. A table with clear information about the forcing data set and types of data that are assimilated and what methods are used to do this. Given that we also know that the strength parameter impacts the thickness it would be good to tabulate P* or equivalent as well. As the paper addresses the timescales of anomalies it would also help to determine the assimilation window length and see if this has an impact.

One reason for requesting the table is when looking at your first figure. You present results but there is not much discussion about the differences that are present in the time series. Are you able to stratify the impact of certain assimilation choices other than whether sea ice data is assimilated.

I would suggest that you may see an impact of the different forcing sets that are used, many are forced with ERA-Interim but MOVE(CORE and G2), GECCO, EDCA, MERRA are not. The ice fields that the atmospheric forcing fields have "seen" will have a impact on the forcing they provide. Differences due to SST relaxation or model parameter choices may also play a role - it would be good to at least see if there are other reasons for the differences other than they include sea ice assimilation or not.

Section 2.3:

It was not entirely clear to me how you were treating draft and SIT differently from the reanalyses when comparing to observations. Did you use the snow cover from reanalyses to compute this when comparing to observations or just disregard for both reanalyses and observations?

Section 3:

Figure 4: some of the scatter diagrams look like the model thickness stops at a particular value e.g. ECDA are you missing some data from thickness categories? Where the draft data do not show a similar relationship to the SIT is there something different about the way snow on ice is treated in the systems? Is this dependent on the assumptions you made about how you compare draft and SIT?

Do you have an understanding of why the largest differences are near the Greenland coast and Canadian Archipelago? is this down to model physics differences?

Section 3.3:

You note that the GloSea systems have shorter timescales than others - is this persistence also linked to mean state? If you have thinner ice on average you may lose it over the summer and it will reduce your persistence.

pg 12, lines 5-6: these sentences were not enitrely clear - would suggest you consider rephrasing to make sure the meaning is clear. do you mean that G2V3, ORAP5, PI-OMAS, TP4 and UR025-4 are just similar and if so why hightlight this group - is it for a particular region of the Arctic. It wasn't totally clear what you wanted to point out here.

pg 12, lines 17-18: do you mean years or months?

Does your wavelet analysis give an indication how robust your findings might be based on the limited time series? If so I think this would be worth commenting on in the text.

Section 4:

pg 17, lines 3-6: You are comparing c_GLORS05-G2V3 - do both of these systems assimilate sea ice data in the same way? You also note that G2V3 and ORAP5 could be considered similar - they may be similiar in some respects but potentially have different forcing G2V3 its not clear to me if G2V3 uses operational analysis from ECMWF rather than ERA-Interim - which may lead to some differences. ORAP5 uses a non-standard value of P* compared to standard LIM2 setup.

Figure 11 shows the reduction in time scale in one system with the application of sea ice assimilation but the change seems somewhat smaller than the differences with the no ice assimilation across the ensemble - would you expect this given you results? where would G2V1 be in your fig 13?

Figure 13 and discussion pg19, lines15-17. I was not convinced that the ice volume anomaly correlated well with the time and length scales - without the GloSea systems it seems less clear.

Technical comments:

Abstract: line 4: intend rather than intent

section 2.2:

pg4, line3: not sure what you mean by ponctual - is this "point" measurements?

pg4, line 14: run by Environment....

section 3.1: pg8, line 2: Don't you mean MERRA rather than MOVE?

section 3.2: pg8, line 22 and pg11, line8: do you mean "error" or "difference" ?

Section 3.4:

the last sentence of the section "..for what a quantification is presented in the conclusions of this study" - the phrasing is somewhat awkward and hard to understand exactly what you mean

---

## Author Comment (AC1) · 19 Dec 2018

The arcticle of "On the time and length scales of the Arctic sea ice thickness anomalies: a study based on fourteen reanalyses" aims at the Arctic sea ice thickness (SIT) which contains considerable uncertainty in the popular 14 reanalyses. They evaluate the reproduced SITs from the reanalyses, and then investigate the e-folding time and length scales of the SIT anomalies. Clearly, these topic and the consequent findings are helpful to deep understanding the SIT and the concerned variability.

We thank the referee for his time and for the detailed revision of our manuscript. We appreciated very much her/his comments, which were all taken into account in the revised version of the manuscript. Below, we answer point-by-point all specific and technical comments.

It is worthwhile mentioning that in order to better incorporate the referees' suggestions, the manuscript's structure has changed. New figures were added, while others were replaced. Thus, all tables, figures, pages and lines referred to in this rebuttal letter are directed to the updated version of the manuscript, unless otherwise stated.

1) Undoubtedly, one conclusion is "reanalyses built with sea ice data assimilation present shorter time and length scales". However, all the reanalyses were only assimilation of sea ice concentration, and the inferred conclusion is not based on the direct comparison of with and without assimilation in a same system frame. The more proofs based on the sea ice concentration will be helpful to increase the rationality on physics. So the counterpart analysis on sea ice concentration shown in Fig. 2 and Fig. 3 will be robust.

We fully understand the referee's comment, since we had the same concern during our analyses: **What are the time and length scales of the Arctic Sea Ice Concentration (SIC)?** Nevertheless, after some tests, we realized that it wouldn't be possible to apply such concepts for this sea ice parameter due to the characteristics of the sea SIC time series. The SIC is not a continuously varying property. For instance, at the region covered by the perennial ice, the SIC is expected to be (nearly) 100% for the all year round. In this case, for instance, the time scale would be "infinity".

For the BGEP mooring's location, notice that the SIC can be nearly constant for several months and it suddenly drops during the summer months (Fig. A, top panel). When calculating anomalies (Fig. A, bottom panel), the values remain near to zero for the months marked by nearly constant SIC. We believe the concepts of time and length scales, as they were explored in the paper, are not applicable for the SIC time series.

2) Section 3.2 illustrates the intercomprison of the reanalyses. The current main features shown by Fig. 5 is not meaningful enough: some reanalyses are very close… In this section, more other information about SIT and its anomaly need to be added.

In the second version of the manuscript, we are bringing new elements for discussion. For instance, we have now compared the reanalyses based on several other parameters and specifications (Figs. 12 and 13). In addition, an extended discussion is presented in Sec. 4.

Firstly, the ensemble mean SITs based on with and without assimilation will be useful (P 15 Line 10: "This suggests that higher length scales are associated with thicker ice"). Furthermore, it will be complementary of the previous knowns in Uotila et al. (2018) and Johnson et al. (2012).

Johnson, M., Proshutinsky A., Aksenov Y., Nguyen A. T., Lindsay R., Haas C., Zhang J., Diansky N., Kwok R., et al.: Evaluation of Arctic sea ice thickness simulated by Arctic Ocean Model

Intercomparison Project models. J. Geophys. Res., 117(C8), C00D31, doi:10.1029/2011JC007257, 2012.

Secondly, the standard deviations of the two ensembled SIT anomalies were not shown before and would be interested to the reader to know the variabilities or the distinguishes about the SIT anomaly in the reanalyses and even considering with and without assimilation.

Fig. B (top row) shows the ensemble mean from all reanalyses (top left), as well as the ensemble mean from all systems with (top center) and without (top left) data assimilation. Overall, there is a good correspondence between the patterns of mean SIT in the three panels, but with slightly thicker ice for the systems with data assimilation, which is mainly distinguished off the northern Greenland coast and around the Canadian islands. Fig. B (bottom row) displays the averaged standard deviation from the three groups of reanalyses (Ensemble, DA and NA). As for the mean fields, there is not a big difference between the three panels. Since these fields don't add much in order to distinguish systems with and without sea ice data assimilation, we preferred to leave this figure here in the discussion.

Nevertheless, we are taking referee's suggestion further in order to better understand the link between Mean State and TS/LS in the new Fig. 13, and its respective discussion in Sec. 4. For this ensemble of reanalyses, where each system considers several different parameters compared to the other systems, it is not clear what is the impact of the variability on the scales, as shown in Fig. D from the rebuttal letter for the second referee.

[Figure]

Fig A. (top) Sea Ice Concentration [%] from C-GLORS05 reanalysis at the location of the BGEP oceanographic mooring. (bottom) Same as top panel, but for the Sea Ice Concentration anomaly [%].

[Figure]

Fig B. (top) Sea ice thickness mean estimated for the (a) entire ensemble, (b) reanalyses with sea ice data assimilation and (c) reanalyses without sea ice data assimilation. (bottom) Same as (top), but for the standard deviation.

3) Figure 2 clearly shows the time scale has been extended from less than 3 months to around 4 months, which is convert with the main finding (P1, Line 9: … data assimilation present shorter time and length scales).

We absolutely agree with referee's comment. In Fig. 2 the reanalyses which do not assimilate data seem to have shorter TS, in contrast with our main finding. Nevertheless, the results that support our main finding is based on the Averaged Weighted Mean (AWM) values, as a representation of all grid cells, while Fig. 2 represents a single grid point.

Fig. 2 should be interpreted with caution. Its main goal is to show that the time scales found for the observations are somehow within the range found for the reanalyses. That's why, Fig. 2 is displayed in Sec. 2.3 ('Methods').

However, we indeed agree that this is an important point and deserves a better clarification in the text (Fig. 11; pg. 20, lines 3-4)

4) The previously compared studies show atmospheric forcing fields essentially drive the results of sea ice simulations (Gerdes and Köberle, 2007; Hunke and Holland, 2007). Can you add some comment or analysis about the impact of the forcing resolutions on the SIT time and length scales?

Gerdes, R., and C. Köberle: Comparison of Arctic sea ice thickness variability in IPCC Climate of the 20th century experiments and in ocean–sea ice hindcasts, J. Geophys. Res., 112, C04S13, doi:10.1029/2006JC003616, 2007.

Hunke, E., and M. Holland: Global atmospheric forcing data for Arctic ice-ocean modeling, J. Geophys. Res., 112, C04S14, doi:10.1029/2006JC003640, 2007.

Even though the atmospheric forcing fields are reported to play a major role in the sea ice simulations, as pointed out by the referee, we could not identify distinguished patterns between the two main sources of atmospheric forcing used by the ensemble of reanalyses: Era-Interim and NCEP/NCAR (Figs. 12c-d). (pg. 20, lines 15-18)

5) The ice draft measurements from submarine have been identified an overall overestimation of +0.29 m (Rothrock and Wensnahan (2007). This dataset also is used in this study. Can you add some comments about these kinds of bias corrections (also to other related observational data) applied here or not.

The Sea Ice CDR already provides the corrected data. For the US submarine case, notice that the files were produced and made available by Dr. Mark Wensnahan http://psc.apl.uw.edu/sea_ice_cdr/Sources/US%20Submarines.html, one of the researchers who identified and reported (Rothrock and Wensnahan, 2007) the biases (pg. 5, line 7).

**Technical issues:**

1) As a basic index calculated by the SIT, it still is not clear how to deal with the conflicts of seawater and ice cover at each grid. For example in Fig. 2, when the observed sea ice is lager 0.1m, but the reanalyses are not all covered by sea ice. It is also not clear at P 15 Line 13 when to calculate the correlation between the two points: how to ensure the same lengths of the SIT time series.

In this work the reanalyses provide the mean sea-ice thickness of each grid cell including open water for the reanalyses (i.e. sivol variable in CMIP6). As adopted by Blanchard-Wrigglesworth and Bitz (2014), only grid points wherein the mean ice thickness at the time of summer minimum is greater than 0.1 m are taken into account.

The draft observations used in Fig. 2 represent the location where the oceanographic mooring (BGEP) was deployed (Krishfield et al., 2013), while the reanalyses' data in the same figure come from the nearest grid point to the BGEP oceanographic mooring.

On pg. 15, line 13, we are discussing about the stability of the length scale over time. In order to perform these analyses, we have used only the two long-term reanalyses (GECCO2 and MOVE-CORE), taking into account an overlapping period of 64 years, from 1948 to 2011 so that both time series have the same length.

2) P3, Line 20: "The original horizontal grids range from 0.25 to 1". It is not correct because the reanalysis of TP4 is regional product with the resolution of 12-16km (also see Xie et al. (2017)).

Xie, J., Bertino, L., Counillon, F., Lisæter, K., and Sakov, P.: Quality assessment of the TOPAZ4 reanalysis in the Arctic over the period 1991-2013, Ocean Sci., 13(1), 123-144, doi:10.5194/os-13-123-2017, 2017.

This information has been corrected in the text. It is worthwhile mentioning that the grid resolution of TOPAZ4 and of all the other reanalyses are now displayed in Table 1.

3) P3, Line 27:" … in details by Chevallier et al. (2017) (their Table 1) and Balmaseda et al. (2015)". It is not correct because they did not include the TP4 product as least so recommend of the reference: Xie et al. (2017) or Uotila et al. (2018).

Corrected.

4) P4, Line 27:" the linear relationship between both parameters given by the hydrostatic equation". It is better if clear to state the used equation or give a reference used.

This statement was tempered in the text. The main point here is the fact that we decided not to use the snow cover from the reanalyses for converting sea ice thickness to sea ice draft. This decision was taken in order to avoid adding uncertainties to the SIT fields. So, when comparing SIT from reanalyses against draft from observations, we estimate the linear relationship between both datasets by means of the coefficient of correlation (R) and the Mean Residual Sum of Squares (MRSS). (pg. 5, lines 25-32)

5) Figure 1 adds the grid lines or labels the year on each panel. It is more convenient to march the statement. So P 11, Line 13 "for instance from 2001 to 2004" looks not suitable, and can be replaced by "for instance from 2002 to 2004".

We have added yearly grid lines to Fig. 1. In the new version of the manuscript we are still referring to the period "2001 to 2004", but indeed the grid lines made the comparison between the panels much easier.

6) P 15 Line 10: "This suggests that higher length scales are associated with thicker ice" looks not so precis. It more likes around the North pole.

The relationship between the mean state and the LS is explored further in the new manuscripts' version (Fig. 13a,b; pgs. 21, lines 1-3). But indeed, the previous statement was more related to the region surrounding the North pole and this is what we tried to say with "… near central Arctic." (pg. 15, line 9 – in the first manuscript).

7) P 18 Line3:" .. scales of the sea ice thickness" replaced by "… scales of the sea ice thickness anomaly."

"anomaly" added to the text.

8) Figure 11 adds a panel to show the difference so that details would be more clear.

Fig. 11 shows the difference G2V3-G2V1, both for time (Fig. 11a) and length (Fig. 11b) scales.

---

## Author Comment (AC2) · 19 Dec 2018

The paper provides a summary of the sea ice thickness anomalies found in the most recent ocean-ice analyses. It considers the impact of the assimilation of sea ice observations within some of these systems to spatial and temporal scales of the anomalies.

In my opinion the paper has lots of potential and is almost there, but at the moment it is missing some extra synthesis/analysis which would make it a really useful reference for the observation and modelling community. The impact of sea ice assimilation was considered (but did not state if any of the models assimilate anything other the concentration, one may nudge the thickness too?); it would be helpful to understand how the impact of other choices in the system may also influence the sea ice thickness anomalies. The sea ice models may have very different methods for modelling thickness. Do they have ice thickness distributions or a single category, does this have an impact? Different atmospheric forcing sets (with different imposed sea ice cover) may influence the local energy balance and in turn affect the ice thickness. I think considering a few other key elements of these systems would make this paper very helpful in guiding the use of reanalyses and their future development (both in terms of the type of data that is used but also the set up of the assimilation systems)

We thank the referee for evaluating our manuscript with such richness of details. We appreciated very much her/his comments, which were all taken into account in the revised version of the manuscript. We have tried to identify the impact of several other reanalyses' choices (10 in total) on the time and length scales of the sea ice thickness anomalies. We did identified other parameters that potentially affect the referred scales, as for instance the air-ice drag coefficient. For the atmospheric forcing the results are not conclusive as detailed in our answer for your second specific comment. None of the reanalyses assimilate sea ice thickness.

It is worthwhile mentioning that in order to better incorporate the referees' suggestions, the manuscript's structure has changed. New figures were added, while others were replaced. Thus, all tables, figures, pages and lines referred to in this rebuttal letter are directed to the updated version of the manuscript, unless otherwise stated.

Specific comments:

The paper provides some details of the reanalyses you have studies but I think it is missing a table summarising the reanlayses - some form of synthesis would be beneficial to the reader, the papers that do tabulate some of this (e.g. Chevallier et al) are not complete for this reanalysis set. A table with clear information about the forcing data set and types of data that are assimilated and what methods are used to do this. Given that we also know that the strength parameter impacts the thickness it would be good to tabulate P* or equivalent as well. As the paper addresses the timescales of anomalies it would also help to determine the assimilation window length and see if this has an impact.

We have now added Table 1 (pg. 4) to the manuscript. The table reproduces the info already presented by Chevalier et al. (2017), but it also brings info for the other four reanalyses not considered by these authors (GECCO2, GloSea5, PIOMAS and TOPAZ4). The following parameters and specifications are included in Table 1: nominal horizontal resolution, ocean-sea ice model, source of atmospheric forcing data, number of ice-thickness categories, EVP or VP dynamics, ice strength parameter (P*) or frictional dissipation coefficient (Cf), air-ice drag coefficient ($C_W$), ocean-ice drag coefficient ($C_A$), source of sea ice data assimilated and method used for assimilating sea ice data.

Since the atmospheric forcing and many other parameters/specifications (see further in this rebuttal letter) do not seem to play a clear role in the time and length scales, and mainly due to the fact that we are working with monthly time series, we don't expect to see large changes in the fields of time and length scales associated to the "assimilation window length". At least, not for this kind of comparison where each product has several different parameter/specification compared to the other systems.

Other parameters could be tested, nevertheless it is not always easy and straightforward to retrieve all information from all reanalyses, as well as to compare products with several varying parameters. Nevertheless, we believe that the list of parameters in Table 1 is comprehensive and **reinforce that time and length scales are mainly driven by the fact of whether or not the reanalyses assimilate sea ice concentration data** (see below).

One reason for requesting the table is when looking at your first figure. You present results but there is not much discussion about the differences that are present in the time series. Are you able to stratify the impact of certain assimilation choices other than whether sea ice data is assimilated.

I would suggest that you may see an impact of the different forcing sets that are used, many are forced with ERA-Interim but MOVE(CORE and G2), GECCO, EDCA, MERRA are not. The ice fields that the atmospheric forcing fields have "seen" will have a impact on the forcing they provide. Differences due to SST relaxation or model parameter choices may also play a role – it would be good to at least see if there are other reasons for the differences other than they include sea ice assimilation or not.

Figs. A and B (attached to this rebuttal letter) show how time and length scales (hereafter, TS and LS) are related to the reanalyses' parameters and specifications displayed in Table 1.

Fig. A shows a comparison of TS and LS against a set of 'uncountable' specifications (sea ice data assimilation, atmospheric forcing, sea ice model, dynamics, EVP or VP dynamics, and sea ice forcing). The results reinforce that TS and LS are indeed linked to the fact of whether the reanalyses assimilate (DA) or not (NA) sea ice data (Fig. A$a,b,i,j$), while the other specifications do not seem to have a large impact on the studied scales (Fig. A$c$–$j$). For the atmospheric forcing case, notice that we can divide the reanalyses into three groups: ERA-Interim (7 out of 14), NCEP/NCAR (3 out of 14) and Others (4 out of 14). So, an effective comparison can be made only between ERA-Interim and NCEP/NCAR. Nevertheless, from NCEP/NCAR 1 system assimilates sea ice data, while 2 other systems do not assimilate sea ice data. This makes it difficult to evaluate the impact of the atmospheric forcing on the studied scales.

Fig. B$g$-$p$ compares TS and LS against the 'countable' reanalyses parameters. A part of the horizontal grid resolution and the air-ice drag coefficient ($C_A$), which show a certain correlation with LS (Fig. B$h$) and TS (Fig. B$m$), respectively, the number of thickness categories, the ice strength parameter ($P^*$), and the ocean-ice drag coefficient ($C_W$) do not show a strong correlation with the studied scales.

In addition, Fig. B$a$–$f$ also compares TS and LS against the mean state (mean Sea Ice Volume; SIV), the interannual of variability (std SIV anomaly), and the Sea Ice Drift (SID). All these parameters show a certain correspondence with the TS and/or LS. The most pronounced is the case Mean Sea Ice Drift $x$ TS shown in panel Fig. B$e$.

The most meaningful results from Figs. A and B are incorporated in the new version of the manuscript (Figs. 12 and 13; Sec. 4)

Section 2.3:

It was not entirely clear to me how you were treating draft and SIT differently from the reanalyses when comparing to observations. Did you use the snow cover from reanalyses to compute this when comparing to observations or just disregard for both reanalyses and observations?

We decided not to use the snow cover from the reanalyses for converting sea ice thickness to sea ice draft. This decision was taken in order to avoid adding uncertainties to the SIT fields. So, when comparing SIT from the reanalyses against Draft from the observations we made use of **two metrics**:

(i) **the correlation coefficient (R)**, as a measure of the **linear correlation** between SIT (reanalyses) and Draft (observations);

(ii) **the Mean Residual Sum of Squares (MRSS)**, as an indicator of whether SIT values from the reanalyses are good **predictors** for Draft observations;

We have improved this info in the text in order to make it clearer to the reader (pg. 5, lines 25-33).

Section 3:

Figure 4: some of the scatter diagrams look like the model thickness stops at a particular value e.g. ECDA are you missing some data from thickness categories? Where the draft data do not show a similar relationship to the SIT is there something different about the way snow on ice is treated in the systems? Is this dependent on the assumptions you made about how you compare draft and SIT?

The data used in this work was previously compiled and published by Chevalier et al. (2017) and Uotila et al. (2018). These authors collected the data as they were made available by the providers. So, it is really unlike that some thickness categories are missing.

In ECDA this feature (Fig. 4) takes place due to the fact that this system is characterized by relatively thin ice. See Fig. C at the end of this rebuttal letter. Notice that the same is also valid for GloSea5 and UR025-4.

Do you have an understanding of why the largest differences are near the Greenland coast and Canadian Archipelago? is this down to model physics differences?

The regions near Greenland coast and Canadian Archipelago are marked by the thickest sea ice over the studied domain (please, see Fig. B from the rebuttal letter to the first reviewer). So, proportionally, the same differences in these regions are amplified when calculating the errors. This is a simple, but interesting, effect that deserves some explanation in the text. Thanks for your observation. (pg. 19, lines 8-10)

Section 3.3:

You note that the GloSea systems have shorter timescales than others - is this persistence also linked to mean state? If you have thinner ice on average you may lose it over the summer and it will reduce your persistence.

Overall, the systems which do assimilate sea ice are marked by a certain correspondence between TS (persistence) and the mean state as shown in Fig. 13a (or Fig. Ba in this rebuttal letter). The same for the LS and mean state (Fig. 13b or Fig. B*b*). Nevertheless, this is an overall observation

for the cloud of points. Looking carefully at the GloSea systems, GloSea5 is indeed characterized by thin ice (second thinner mean state of the ensemble). On the other hand, GloSea5-G05 has thicker ice compared to GloSea5. Compared to the ensemble of the reanalyses which do assimilate sea ice, the GloSea5-G05 presents an intermediate mean state. In Sec. 4 (pg. 22, lines 11-20), we raise some aspects that could potentially impact the scales of the GloSea systems.

pg 12, lines 5-6: these sentences were not enitrely clear - would suggest you consider rephrasing to make sure the meaning is clear. do you mean that G2V3, ORAP5, PIOMAS, TP4 and UR025-4 are just similar and if so why hightlight this group - is it for a particular region of the Arctic. It wasn't totally clear what you wanted to point out here.

We agree that this sentence was confusing, and we have reconsidered it. (pg. 14, lines 5-6)

pg 12, lines 17-18: do you mean years or months?

We meant months. Corrected in the text.

Does your wavelet analysis give an indication how robust your findings might be based on the limited time series? If so I think this would be worth commenting on in the text.

The "cone of influence" shown in Fig. 7c,g (cross-hatched areas) highlights the region of the wavelet spectrum where edge effects become important due to the length of the time series. In this region of the spectrum, the results should be interpreted carefully, since the time series is short for solving some periods (y-axis) in some specific time spans (x-axis). This information is found in the Fig. 7's caption.

Section 4:

pg 17, lines 3-6: You are comparing c_GLORS05-G2V3 - do both of these systems assimilate sea ice data in the same way? You also note that G2V3 and ORAP5 could be considered similar – they may be similiar in some respects but potentially have different forcing G2V3 its not clear to me if G2V3 uses operational analysis from ECMWF rather than ERA-Interim - which may lead to some differences. ORAP5 uses a non-standard value of P* compared to standard LIM2 setup.

Indeed, C-GLORS05 and G2V3 assimilate sea ice data in a different way (Table 1), and we agree with referee's comment that this may be a reason for the relatively high RMSE between these two systems. The three referred reanalyses (C-GLORS05, G2V3 and ORAP5) apply the same atmospheric forcing (ERA-Interim). C-GLORS05 and G2V3 assume P* = 2.0 x 10$^4$, while ORAP5 uses P* = 1.5 x 10$^4$.

Based on referee's observations, we brought this discussion to the text (pg. 18, lines 5-12).

Figure 11 shows the reduction in time scale in one system with the application of sea ice assimilation but the change seems somewhat smaller than the differences with the no ice assimilation across the ensemble - would you expect this given you results? where would G2V1 be in your fig 13?

Following referee's 1 suggestion, the new version of Fig. 11 shows the TS and LS differences between the systems (G2V3 – G2V1). Fig. 11 suggests that the fact that time and length scales are shorter for systems with data assimilation is valid in terms of pan-Arctic averages, but not necessarily at every grid cell. The pan-Arctic AWM-TS and AWM-LS from G2V3 are 5 months

and 728.2 km, respectively. Without sea ice data assimilation (G2V1), the AWM-TS and AWM-LS increase to 5.5 months and 745.3 km, respectively. For this case, TS increases by 10% from G2V3 to G2V1, though the LS scale isn't strongly impacted (pg. 20, lines 1-6). In Sec. 4, we are discussing that TS is more sensitive to the systems' parameters and specifications.

Fig. 14 (empty red circle; highlighted in the gray rectangle) displays where G2V1 takes place in the diagram TS $x$ LS.

Figure 13 and discussion pg19, lines 15-17. I was not convinced that the ice volume anomaly correlated well with the time and length scales - without the GloSea systems it seems less clear.

We agree that in the way the figure was plotted, it wasn't easy to identify the correlation between mean state and the scales (TS and LS). So, we have now plotted two different diagrams: Mean SIV $x$ TS (Fig. 13a) and Mean SIV $x$ LS (Fig. 13b). Besides the spread among the points and the relatively weak correlations (which are not significant at the 5% level), an increase in the mean SIV generally leads to longer time and length scales, taking into account that each reanalysis has different sets of specifications and parameters (pg. 21-22).

Technical comments:

Abstract: line 4: intend rather than intent

Corrected.

section 2.2:

pg4, line3: not sure what you mean by ponctual - is this "point" measurements?

Corrected.

pg4, line 14: run by Environment.…

Corrected.

section 3.1: pg8, line 2: Don't you mean MERRA rather than MOVE?

Indeed, we meant MERRA. Corrected in the new version of the manuscript.

section 3.2: pg8, line 22 and pg11, line8: do you mean "error" or "difference" ?

We meant error (RMSE). Clarified in the text.

Section 3.4:

the last sentence of the section "..for what a quantification is presented in the conclusions of this study" - the phrasing is somewhat awkward and hard to understand exactly what you mean

Corrected.

[Figure]

Fig. A: Histograms showing how the AWM-TS (left panels) and AWM-LS (right panels) are related to other "uncountable" reanalyses' parameters and/or specifications, as follows: (a-b) whether or not the system assimilates sea ice data; (c-d) the source of atmospheric forcing data; (e-f) the used sea ice model; (g-h) the used dynamics (Viscous-Plastic or Elastic-Viscous-Plasctic) to account for ice-ice interactions that control ice deformation'; and (i-j) the source of assimilated sea ice concentration data.

[Figure]

Fig. B: Scatter plots showing how the AWM-TS [months] (first and third columns) and AWM-LS [km] (second and fourth columns) are related to other reanalyses' parameters, as follows: (a-b) Mean Sea Ice Volume [m] (SIV); (c-d) standard-deviation of the SIV Anomaly [m]; (e-f) Mean Sea Ice Drift [m/s] (SID); (g-h) grid resolution [degrees]; (i-j) number of thickness categories for discretization of ice thickness; (k-l) ice strength parameter (P*) [N/m] for the ice strength formulation following Hibler (1979); (m-n) air-ice drag coefficient [$10^{-3}$] ($C_A$); and (o-p) ocean-ice drag coefficient [$10^{-3}$] ($C_W$).

[Figure]

Fig. C: Radar plots for the Averaged-Weighted-Mean Sea Ice Thickness. Every color represents a different year from 1993 to 2007.